# The Role of Microorganisms in the Nucleation of Carbonates, Environmental Implications and Applications

Ana Robles-Fernández [1,2], Camila Areias [1], Daniele Daffonchio [3], Volker C. Vahrenkamp [2] and Mónica Sánchez-Román [1,*]

1 Earth Sciences Department, Vrije Universiteit Amsterdam, 1081 HV Amsterdam, The Netherlands
2 Ali al Naimi Petroleum Engineering Research Center (ANPERC), King Abdullah University of Science and Technology (KAUST), Thuwal 23955-6900, Saudi Arabia
3 Red Sea Research Center (RSRC), King Abdullah University of Science and Technology (KAUST), Thuwal 23955-6900, Saudi Arabia
* Correspondence: m.sanchezroman@vu.nl

**Abstract:** Microbially induced carbonate precipitation (MICP) is an important process in the synthesis of carbonate minerals, and thus, it is widely explored as a novel approach with potential for many technological applications. However, the processes and mechanisms involved in carbonate mineral formation in the presence of microbes are not yet fully understood. This review covers the current knowledge regarding the role of microbial cells and metabolic products (e.g., extracellular polymeric substances, proteins and amino acids) on the adsorption of divalent metals, adsorption of ionic species and as templates for crystal nucleation. Moreover, they can play a role in the mineral precipitation, size, morphology and lattice. By understanding how microbes and their metabolic products promote suitable physicochemical conditions (pH, Mg/Ca ratio and free $CO_3^{2-}$ ions) to induce carbonate nucleation and precipitation, the manipulation of the final mineral precipitates could be a reality for (geo)biotechnological approaches. The applications and implications of biogenic carbonates in areas such as geology and engineering are presented and discussed in this review, with a major focus on biotechnology.

**Keywords:** carbonates; microorganisms; bacteria; biomineralization; organics; diagenesis; EPS; bioremediation

## 1. Introduction

Carbonate sediments are abundant as shallow-water limestones throughout the Phanerozoic and as both shallow-water limestones and deep pelagic ooze since the end of the Triassic period [1]. Classically, carbonates have been thought to be formed via abiotic chemical diagenetic processes consisting of four key steps: (i) the increase in the calcium concentration, (ii) a high concentration of dissolved inorganic carbon, (iii) an increase in the water/seawater pH (the "alkalinity engine") and (iv) the availability of nucleation sites [2]. Nonetheless, in the deposition processes of carbonates, such as calcite [3], aragonite [4], vaterite [5], Mg-calcite [6] or dolomite [7], and even in the precipitation of other minerals, such as phosphates [8–10], oxides [11,12], sulfur/sulphates [13,14] and/or silicates [15], microorganisms have been frequently considered to play a key role in their mineral nucleation and precipitation, performing a process known as biological mineralization.

Biomineralization is defined as the ability of certain organisms to form minerals by favoring their chemical precipitation as a side consequence of their metabolism [16–22]. Many organisms are able to biomineralize, from microbes to superior organisms, such as mollusks, corals, crustaceans or some species from the Hymenoptera order that are able to create exoskeletons, such as shells, and endoskeletons, including mammal and bird's bones [23–29]. Bacterial cells can offer nucleation sites on their cell wall surface to the ions implicated, and, additionally, such ions can interact with molecules excreted

by the cells present in the micro-environment, such as extracellular polymeric substances (EPS) or even organic acids [30]. This process can also be subdivided into two types: (i) biologically induced mineralization (BIM), which occurs when an organism modifies its local environment creating conditions such that there is extracellular chemical precipitation of mineral phases; (ii) biologically controlled mineralization (BCM) in which, as opposed to BIM, a genetic control of the biomineralization is exerted by the microbial cells [31]. One of the most studied examples of biomineralization is the precipitation of calcium carbonate minerals, with great influence on microbial metabolism [30,31].

Microbial mats, which are composed of thick layered films on microorganisms, mainly bacteria and archaea [32], or only bacteria [33], are well known as biomineralizing organisms. The study of microbial mats and how they influenced the early Earth's environmental conditions is fundamental, as it provides a record of the past climate changes on Earth, extending back to the Precambrian [34–36].

Bacteria and archaea may induce and control the precipitation of carbonate minerals at the nano- and micrometric scale. The different products of their metabolism, such as proteins (especially enzymes), polysaccharides and other EPS, and the charged groups embedded in their microbial cell walls, by interacting with anions and cations, affect the crystal properties and chemical composition of the minerals. With respect to chemical deposition, the microbial activity can enhance the reaction rates, providing a favorable way to accumulate minerals to be recovered for applied purposes, and may modify carbonate mineral structures, such as $CaCO_3$, conferring properties different from those originating from the sole chemical precipitation, as illustrated in Figure 1. The elucidation of the undergoing molecular mechanisms might be very advantageous for generating products for the industrial application of carbonates, which could also be scalable and economically profitable [2]. Photosynthetic microorganisms (algae and cyanobacteria), dissimilatory sulfate-reducing bacteria, heterotrophic microorganisms that produce and use organic acids, and (macro-)microorganisms involved in the nitrogen cycle either performing ammonification, amino acid metabolism, or hydrolysis of urea are being studied for microbially induced carbonate precipitation (MICP) [30]. Nonetheless, not only can such microorganisms precipitate carbonates but also higher organisms such as corals which precipitate carbonates as mechanical support in coral reefs or shell-producing animals.

Just to list a few of the potential biotechnological applications of MICP, it has been studied and demonstrated that MICP, in combination with other biomineralization processes, such as enzymatically induced carbonate precipitation (EICP), could be useful for the control of dust in polluted cities [37] as well as in the consolidation of ground and sands that have been affected by rainfall erosion [38]. MICP and enzymatic treatments have also been considered to trap heavy metals in carbonate crystalline structures for the bioremediation of metal-contaminated soils [39]. In addition, enzymatic treatments have been studied to enhance sand biocementation in calcareous beaches by immobilizing ions in the sand pores [40]. Enzymatic activities have also been proposed as tools to improve the sample processing of different analytical tests [41] and for many more applications that are further explored in Section 5.2. Carbonates are important resources as well, since limestone and dolomite are reservoir rocks that contain approximately 38% of the planet's oil and gas reserves [42], and they are important freshwater aquifers in many regions [43]. Therefore, carbonates are essential as resources of energy and water for the future [44]. Carbonates are also considered potential geological deposits for the injection of $CO_2$, removing it from the atmosphere and storing it underground after mineralization processes, thus contributing to reducing greenhouse gasses and the effects of climate change [45].

The aim of this study was to examine the biological mechanisms involved in carbonate precipitation as well as to provide an overview of recently proposed applications of biologically synthesized carbonates in different areas of biotechnology and sustainability.

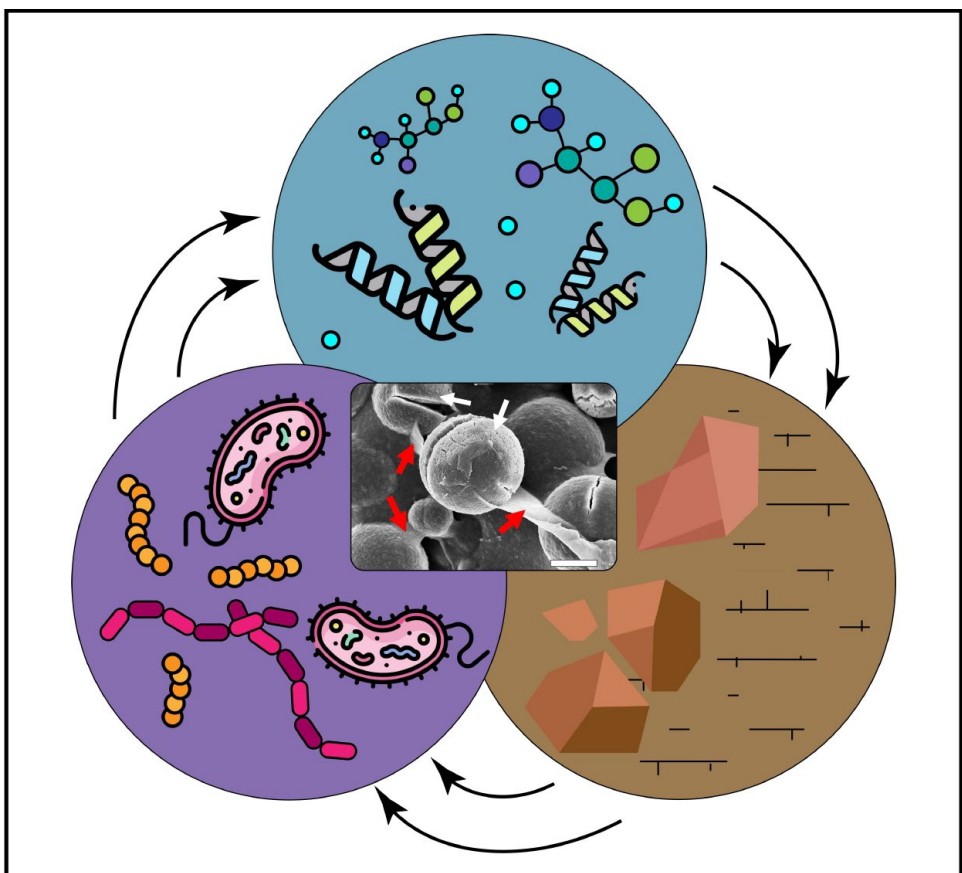

**Figure 1.** Circle diagram showing the biogenic formation of carbonate minerals. The compendium of minerals found in the natural environments (brown circle), microorganisms capable of forming carbonate minerals (purple circle) and the biochemistry of the process at a nano- and micrometric scale involving the biomacromolecules produced by the microorganisms that interfere in the nucleation and mineral growth process (green circle), resulting in the biogenic creation of minerals associated to biological processes (center). The SEM image in the center represents rounded spherical crystals made of Ca-Mg carbonate (white arrows) bonded together by EPS (red arrows). Scale bar: 10 μm.

## 2. Biopolymers, Organic Compounds Relevant for Carbonate Precipitation

The present study focused on the organic polymers (biopolymers) produced by microorganisms, such as proteins, enzymes, nucleic acids, carbohydrates and/or extracellular polymeric substances (EPS), that present a huge impact on many biological processes which can be industrially scaled, including mineral precipitation, especially in carbonate minerals, such as calcite, aragonite, vaterite, Mg-calcite and/or dolomite [46]. They are biomacromolecules, large molecules assembled from monomers linked together in variable spatial structures with different physicochemical properties and present in all organisms [47]. They have a role in most biological processes, with a large variety of functions. Proteins are made up of amino acid monomers, characterized by a backbone of amino and carboxyl functional groups [47]. Proteins can fold into different spatial conformations. The formation of 3D structures by the interaction between different amino acids and the surrounding environment creates pockets with different chemical environments [47]. The chemical conditions created in such catalytic pockets are the key to developing their wide range of functions for interacting with other pockets from other proteins, nucleic acids, small molecules or even water molecules [48]. Some examples of small molecules that can interact with them are substrates, metal ions, prosthetic groups or cofactors [48]. Most of the three-dimensional structural conformations are generally reversible. Thus, proteins can switch between alternative conformational structures in order to modify the specificity of interactions with different molecules [47,48].

Among all the different types of proteins and the wide range of functions proteins are able to perform, there is a special group of them, the enzymes. Enzymatic catalysts, under physiological conditions of temperature, pH and pressure of the cells are capable of carrying out chemical reactions at much higher rates than the same reactions performed only by the chemical components involved, without altering the chemical equilibrium [49], and lowering the Gibbs free energy by acting as transferases, oxidoreductases, hydrolases, lyases, isomerases, ligases and translocases [50,51].

Apart from these enzymes, structural proteins are found in all cell types. A large number of them are well located in cell membranes, embedded in the lipids layer. This fact is remarkable due to the different ranges of charge of the proteins present, which depends on the residues/amino acids that they are composed of, something that enables these proteins to attract different anions or cations to the cell surface [49,52]. The attraction of these molecules has special interest in biological diagenetic processes, since it allows to oversaturate the precipitating solutions with the ionic species needed for mineral formation [53].

Proteins and enzymes are not the only biomolecules considered to be involved in the carbonate crystal nucleation and in the precipitation processes. Polysaccharides and carbohydrates, the molecules of which consist of repetitions of a number of sugar molecules bonded together by covalent interactions, have also been related to biomineralization processes [54–60]. To each of all the different kinds of polysaccharides found in living cells, the ones considered to be enrolled in the biological formation of different minerals are the EPS produced and excreted by a large number of bacteria and other organisms. The EPS are a high mixture of high-molecular-weight polymers, mainly composed of polysaccharides with some nucleic acids, lipids, proteins and other molecules, such as humic acids, pyruvate, acetate or succinate [61].

The EPS are the major components of biofilms (up to 90%) [61], and they carry out functions such as the aggregation of cells and the attachment of cells to several surfaces [62]. The EPS also concentrate different substances (for instance, nutrients or ionic species) that act as reactants in chemical reactions outside the cells. The EPS also can be used as transporters of chemical messenger molecules between different cells [62]. Their utility has also been related to the protection from external damaging and toxic agents [63], as shown in Figure 2. Genetic material may even be exchanged between microorganisms through the EPS matrix [64]. Additionally, EPS have the capability of changing the Mg/Ca ion ratio [65]. The EPS favor the formation of Mg-rich carbonate minerals because of their ability to dehydrate bonded $Mg^{2+}$ ions, so they can be incorporated into the carbonate crystalline structure [66–73]. It is thought that divalent ion sequestration happens due to the high number of negatively charged groups found in EPS, such as carboxylic groups [74]. The mechanism of binding metals mechanism and, thus, the ability to change the metals/ions ratio has been described both in eukaryotic and in bacterial and archaea cells [75].

Biopolymers, such as proteins, enzymes and polysaccharides, are produced and can be secreted out of the cell, and they can participate passively or actively in the formation of different minerals, such as carbonates [3], phosphates [8–10], oxides [12], sulfur/sulfates [13] and/or silicates [15].

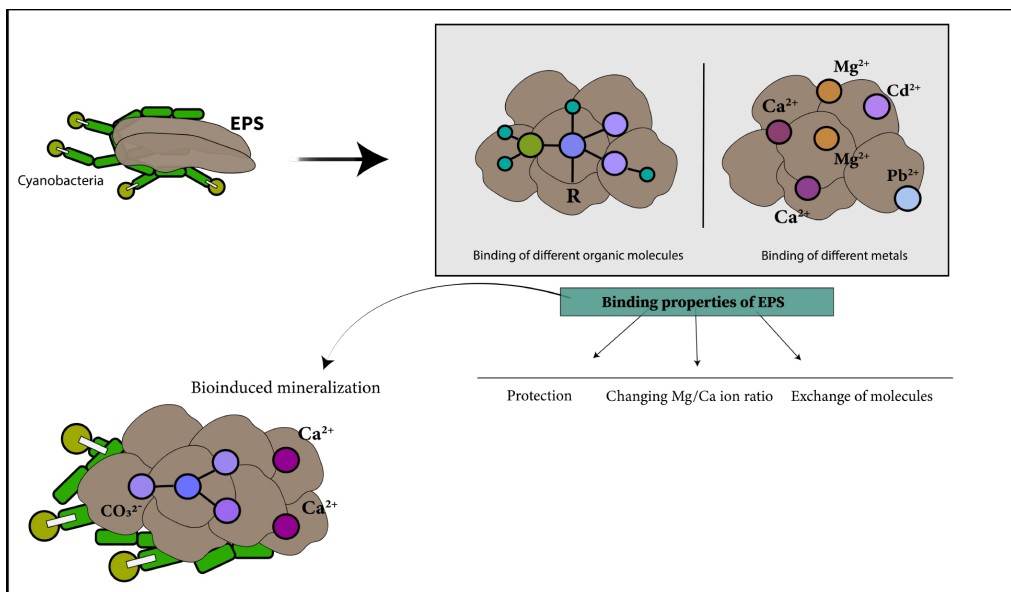

**Figure 2.** Schematic representation of EPS properties that contribute to the formation of biogenic carbonates. Binding properties of EPS are related to biological functions, such as protection for the producer microorganisms, ability to change the Mg/Ca ion ratio of the surrounding environment and the capacity to exchange molecules between the pericellular space and the microorganisms as well as between the microorganisms embedded in the EPS matrix. Its binding properties directly influence the carbonate mineral precipitation by the concentration of the metal ions and reactive molecules involved in the nucleation process [75–79].

## 3. Types of Microbial Biomineralization

Microorganisms are nowadays very powerful tools in the biotechnological industry, and their implication in physico-chemical processes has been described and exploited for a long time [80]. Thus, humans can take advantage of their metabolic products by controlling their metabolic pathways and steering such pathways towards the production of specific metabolites and industrial applications [80–82].

Biomineralization describes a process in which living beings are able to produce minerals. These processes are usually ecofriendly, industrially scalable and may be controllable by manipulating physicochemical parameters such as temperature, pH, ion concentration and the time of incubation and chemical composition of the growth medium [83], avoiding the toxicity associated with physiochemical synthesis [84]. Additionally, this process presents higher biocompatibility rates when used in human health approaches [81,85]. Mineral biosynthesis can be categorized into [16,31,86–90]:

- Biologically influenced mineralization (organomineralization): Passive/indirect microbial biomineralization or biological mineralization. Minerals precipitate by the presence of microorganisms in the environment and the molecules and compounds they may have in their cell wall and/or secreted substances (e.g., EPS), acting as passive nucleation sites. An organic matrix/template enhances the crystal nucleation and precipitation. In this case, living (micro)organisms are not necessarily involved;

- Biologically induced mineralization (BIM): Active/direct microbial biomineralization or biological mineralization. Minerals precipitate as a consequence of the microbial metabolisms (i.e., sulfate reduction, aerobic respiration, methanogenesis, sulfur oxidation and photosynthesis). The metabolic activity of living microbial cells changes the physicochemical conditions (e.g., pH, Mg/Ca and alkalinity) of their surroundings, resulting in the inorganic mineral precipitation and mineral crystal growth. The microbial activity may also affect the mineral precipitate morphology, texture and chemical composition. The parameters modified may not offer any fitness advantage to the (micro)organisms involved but to the environment where they live;

- Biologically controlled mineralization (BCM): The genome of the (micro)organism(s) involved encodes for enzymes or metabolisms specifically evolved to act on the nucleation, precipitation and growth of the mineral crystals, and their synthesis is usually related to the survival of the producing organisms. The acting (micro)organisms are those benefiting from the biomineralization. This is an example of living organisms such as mollusks, which build shells for protection [24], or magnetotactic bacteria and their magnetosomes [11,91,92]. Not only isolated genes, but even entire gene clusters for biomineralization have been sequenced in microorganisms such as *Bacillus subtilis* [93].

Calcium carbonates minerals, in particular, are an example of BIM, because their biological formation is widely observed in nature [94–96]. BIM and, more concretely, MICP can occur as a by-product of some common microbial metabolic activities, from the secondary interactions between metabolic processes producing carbonate species and the surrounding microenvironment [97]. The attraction of the molecules (e.g., $Ca^{2+}$ ions or $CO_3^{2-}$) included in the crystalline structures of different minerals has a special interest in biological diagenetic processes [98], since the saturation of the immediate environment with these molecules is key to triggering the mineral formation. From a physicochemical point of view, the induction of biomineralization is caused by two main factors: (i) carbonate alkalinity and (ii) the availability of free calcium in the solution. These two factors are parametrized in the saturation index, defined as:

$$SI = \log (IAP/K_{sp}) \tag{1}$$

In Equation (1), IAP refers to the ion activity product, which is the activity product (the effective concentration) of the $Ca^{2+}$ and $CO_3^{2-}$ ions, and $K_{sp}$ is the solubility product of the mineral. Regarding the equation, the higher IAP is relative to $K_{sp}$, the saturation index may become more positive (favoring a progressive saturation of the solution), and the eventual precipitation of the mineral might be enhanced. The precipitation is also determined by the chemical equilibrium among all the ions involved in obtaining $CO_3^{2-}$, as shown in Reaction (2):

$$H_2CO_3 \Leftrightarrow HCO_3^- + H^+ \Leftrightarrow CO_3^{2-} + 2H^+ \tag{2}$$

This equilibrium strongly depends on the pH of the medium, and so higher pH values (alkaline conditions) will lead to greater concentrations of $CO_3^{2-}$ in the medium because of a decrease in the free $H^+$ around. Various processes can increase the carbonate alkalinity needed for the precipitation, all of these being processes involved in what is known as the "alkalinity engine" [22,94]. The alkalinity engine can be driven by microorganisms that are able to change their surrounding environment in order to improve the concentration of $CO_3^{2-}$ ions by intervening in this equilibrium via different metabolic pathways. The production of different sorts of biopolymers highlights one of the most important microbial metabolic pathways involved in the MICP process. For instance, it is thought that the enzyme urease and many ureolytic microorganisms may induce the precipitation of calcium carbonates [99,100] as well as lysozyme or even those enzymes involved in the $CO_2$ fixation in photosynthesis, such as carbonic anhydrase (CA) [100,101]. Apart from the protein enzyme catalyzers, the production and excretion of polysaccharidic EPS of different chemical natures might change the close-cell-wall medium, promoting a change in conditions of the cell surrounding environment. The precipitation could also be affected by active–passive changes in charge and composition of the external structure of different charged proteins embedded in cell walls [102]. The $Ca^{2+}$ supersaturation is also a key factor, since calcium is a very important second messenger in many cellular processes [103]. Therefore, it is crucial for cells to be able to regulate their calcium concentration inside and outside of the cytoplasm. Nonetheless, not only the supersaturation performed by these bacteria leads to the biomineralization process but the high pH in the external immediate environment is also important. The pH conditions necessary for the precipitation of $CaCO_3$ are created within the EPS and the S-layer and additionally because of the action of carbonic

anhydrase (CA). CA is an enzyme that consumes $H^+$ and produces $OH^-$ groups, secreted into the medium. In addition to this, the photosynthetic electron transport (PET) and the transport system of $H^+/Ca^{2+}$ antiport transport systems contribute to the perfect conditions for MICP [22].

For example, the microbial mats grown on the surfaces of solid substrates are mostly inhabited by autotrophs, which use inorganic carbon as a main source of energy. They are usually vertically organized according to functional groups, which are interacting in close relation along an organic matrix [32,33]. The distribution of microbial types along the vertical thickness of a microbial mat usually follows the aerobic to anaerobic gradient occurring from the surface [104] to bottom of the mat, whereas (i) cyanobacteria, as oxygenic phototrophs, are followed by (ii) aerobic heterotrophs; (iii) anoxygenic photosynthetic bacteria (generally represented by nonsulfur green bacteria) and green–purple sulfur bacteria; (iv) sulfate reducers (performing dissimilatory sulfate reduction); and, finally, (v) methanogens [105], as simplified in Figure 3. The main bacterial groups found in microbial mats are cyanobacteria, photosynthetic organisms, as well as a smaller group composed of eukaryotic algae, diatoms and phototrophic. These microorganisms, organized vertically, create physicochemical gradients formed and maintained by the activities of the different components of this layered microbiome that create microenvironments where the different metabolic functions of each microbial group can be performed. Cyanobacteria inhabit the top layers of the mat where they carry out photosynthesis [106]. They colonize numerous environments, from terrestrial to marine, even in the most extreme environments, such as deserts or frozen soils (i.e., permafrost). Cyanobacteria in the phytoplankton are responsible for half of the global photosynthetic processes [106], playing a primary production role in the carbon cycle. Taking their carbon production into account, and since they are the oldest fossil records found, cyanobacteria are probably responsible for our current redox conditions of the Earth's atmosphere. The special characteristic of cyanobacteria and eukaryotic algae is that they present a carbon concentrating mechanism (CCM), a metabolic system that allows the cells to enrich in $CO_2$ the small environments inside and outside the cells [107]. Such accumulation occurs where the enzyme Rubisco is present, so $CO_2$ can be fixed into organic carbon molecules. The carbon capture pathways depend on gas diffusion through the action of different membrane transporters [106]. The constant flux of $CO_2$–$HCO_3^-$ and the hydration of these two species due to the CA action determines the $HCO_3^-$ supersaturation in their surrounding environment.

Not only are cyanobacteria very good precursors of the $CaCO_3$ precipitation inside microbial mats, but the role of the sulfate cycle has also been demonstrated to be crucial, mainly by the action of sulfate-reducing bacteria (SRB) through the dissimilatory sulfate reduction [108,109]. SRB reduce sulfate to sulfide while oxidizing organic carbon to bicarbonate [110]. During this reaction, the pH and the $CO_2$–$HCO_3^-$ saturation state is increased [33]. In addition, it has been demonstrated that the cell walls of SRB are able to concentrate higher amounts of $Ca^{2+}$ ions by the formation of EPS, which is secreted to the outside of the cells [111,112]. Thus, the conditions for $CaCO_3$ precipitation are favored.

One of the most important, most studied and simplest examples of MICP processes is that performed by ureolytic microorganisms. Ureolytic microorganisms are able to hydrolyze urea with the urease enzyme that irreversibly transforms 1 mol of urea into 2 mol of $NH_4^+$ and 1 mol of $CO_3^{2-}$ ions, following the Reactions (3)–(7) [97,113,114]:

$$CO(NH_2)_2 + H_2O \rightarrow 2NH_3 + CO_2 \tag{3}$$

$$CO_2 + H_2O \rightarrow H_2CO_3 \tag{4}$$

$$2NH_3 + 2H_2O \rightarrow 2NH_4^+ + 2OH^- \tag{5}$$

$$2OH^- + H_2CO_3 \rightarrow CO_3^{2-} + 2H_2O \tag{6}$$

$$CO_3^{2-} + Ca^{2+} \rightarrow CaCO_3 \text{ (precipitates)} \tag{7}$$

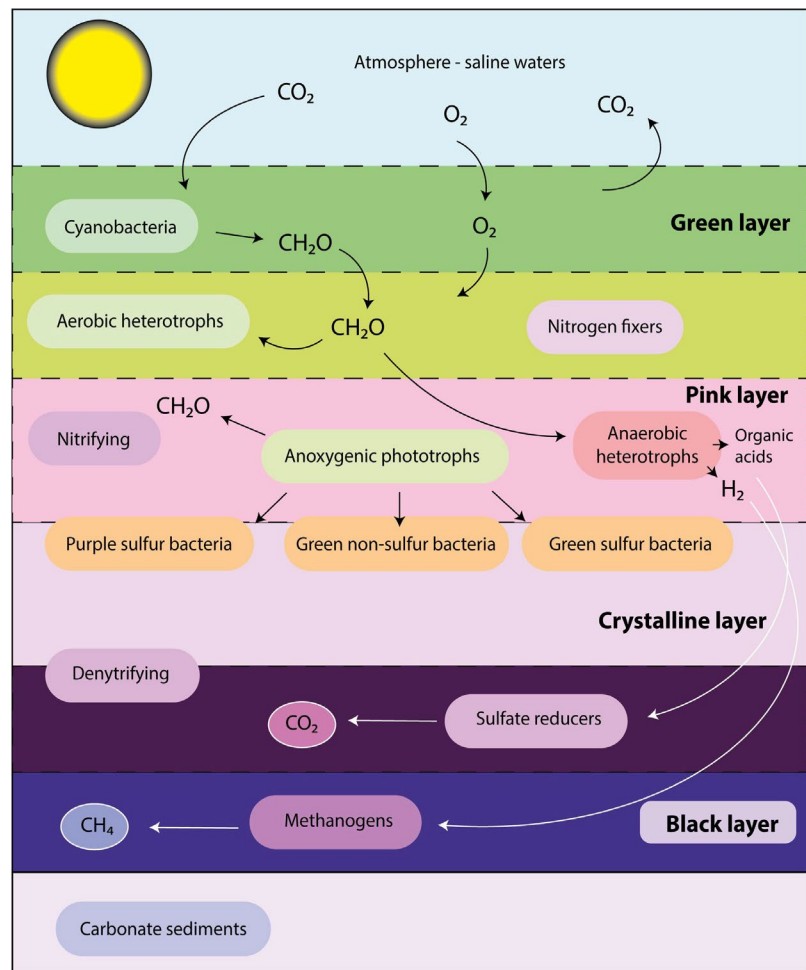

**Figure 3.** Schematic representation of the vertically stratified organization of microbial mats and the simplified carbon cycle that takes place on them. Adapted from Prieto-Barajas et al. (2018) [104].

The increase in $CO_3^{2-}$ ions and the increase in pH for the alkalinity engine are two of the main factors that affect the calcium carbonate precipitation. Therefore, it could be assumed that the ureolytic activity can directly affect the precipitation of carbonates, considering that high concentrations of metal cations ($Ca^{2+}$, $Mg^{2+}$, $Fe^{2+}$, $Mn^{2+}$, $Sr^{2+}$, etc.) should be present in the medium. Additionally, the nucleation sites offered by the negatively charged residues in the cell wall and S-layer are required [53,115]. It has been experimentally demonstrated that a higher concentration of $Ca^{2+}$ combined with a higher rate of urease (higher urea hydrolysis rates) leads to a higher quantity of $CaCO_3$ precipitated [116]. Moreover, it has also been reported that the rate of precipitated $CaCO_3$ is proportional to the cellular growth, because there is a direct relation between the increase in insoluble $Ca^{2+}$ in the solution and the increase in the number of cells in a growth medium [117]. Cell growth and the insoluble $Ca^{2+}$ in the medium have similar increasing trends as well as ammonia ($NH_3$) and the pH.

Urease enzyme presents a high Michaelis–Menten constant ($K_m$) and, thus, a high affinity for urea when the pH rises to 8.0; some studies have proved that urease activity remains high at pH values between 8.3 and 9.0 [117]. The pH range of urease activity is very similar to the pH range at which $CaCO_3$ precipitation is favored. As shown in Equations (3)–(6), it can be assumed that the increase in pH associated with the production of $NH_4^+$ ions by the urease activity might be controlled by the respiration processes of the microbial metabolism, and the urease activity releases both $CO_2$ as $HCO_3^-$ and $CO_3^{2-}$. Thus, the microbial metabolism sustains the optimal conditions for $CaCO_3$ precipitation in time, and so the rates of precipitation are greater. While many microorganisms are considered as

ureolytic, in most of the occasions the ones used in experimental approaches are strains of *Bacillus pasteurii*, (also known with the new taxonomy as *Sporosarcina pasteurii*) [118–122], since it is known as a nonpathogenic ureolytic microorganism with higher rates of urease activity (and high $K_m$) in soil environments, much more efficient in ureolytic activity than other prokaryotes and eukaryotes [123]. Figure 4 exemplifies how the microbial growth and metabolism, combined with the changes that microorganisms produce in the extracellular space, influence certain physico-chemical parameters (metal concentration, pH, etc.) that affect to the production of carbonate minerals.

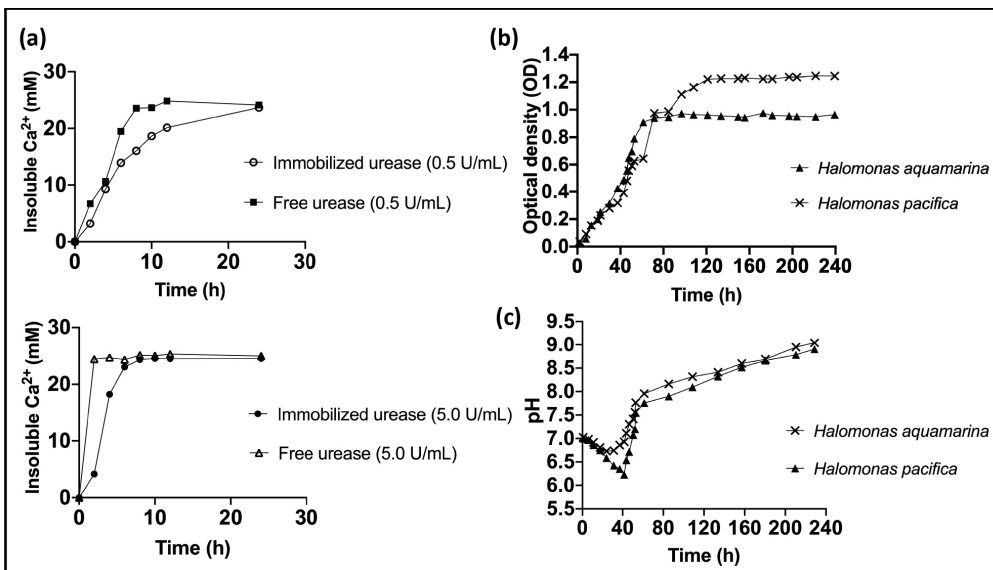

**Figure 4.** (**a**) Different concentrations of the enzyme affected proportionally the concentration of mineral synthesized. The biological accessibility of the enzyme, comparing free *B. pasteurii*'s urease expressed by a transformed *E. coli* strain (free urease) and urease partially purified and immobilized in polyurethane (PHU) foam (immobilized urease), also affected the concentration of insoluble $Ca^{2+}$ (adapted from Bachmeier et al. (2002) [124]). (**b**) Microbial growth interpreted as optical density correlated directly to higher concentrations of catalytic enzymes that influence the amount of biominerals synthesized in the biomineralization model of species *Halomonas* (adapted from Sanchez-Roman et al. (2007) [65]). (**c**) Changes in pH from neutral to alkaline associated to the growth and biomineralization performance of *Halomonas* (adapted from Sanchez-Roman et al. (2007) [65]).

The possibility of inducing carbonate precipitation with enzymes rather than microorganisms in a process known as enzymatically induced carbonate precipitation (EICP) has also been suggested as an appealing option. Even though the precipitation mediated by microorganisms has always gained most of the attention, the use of the free enzyme offers some advantages, such as direct enzymatic catalysis, without the need of growing and storing bacterial cells [125]. Urease production can be cheaper than producing the biomass of ureolytic bacteria when it is produced and extracted from plants. The urease produced by plants has a bigger size (12 μm) compared to that produced by most bacteria (0.5 to 5 μm) [125]. Such difference in size might make a significant difference when increasing the efficiency in biotechnological applications. Additionally, the activity is also distinguishable. In the case of MICP, the bacterial metabolism and, therefore, the enzymatic activity can be inhibited by high concentrations of urea (ureolytic microorganisms are resistant to their own product up to a certain concentration limit) and by the bacterial cell mineralization and encapsulation into the crystal mineral, which would reduce the nutrient flow and then lead to the bacterial cell death. In the case of enzymes, they can be affected by inhibitors as well, but it can be controlled by the use of enzymes immobilized in matrices [125]. Lastly, enzymes are short living, and, thus, once their catalytic activity has been

performed, they biodegrade without causing any accumulation/contamination/biofilm formation processes [125,126].

Apart from ureolytic *Sporosarcina pasteurii*, some Gram-negative aerobic microbial strains are capable of using amino acids as their energy source in order to initiate the biomineralization of calcium carbonates. One example is *Myxococcus xanthus*, which has been demonstrated to induce calcium carbonate precipitation in media containing calcium acetate [127], linking the whole nitrogen cycle and metabolism to biomineralization processes. Other examples of microorganisms determining calcium carbonate precipitation are those involved in the denitrification process, such as *Alcaligenes*, *Denitro bacillus*, *Thiobacillus*, *Spirillum*, *Micrococcus* and *Pseudomonas denitrificans*, among others [113,128,129].

Several microbiome-dependent environmental parameters influencing the biomineralization process are crucial as well. MICP processes are controlled by several environmental factors, such that the interaction with the ambient surroundings of the microorganisms/enzymes is crucial. One of the main factors influencing the process is temperature, because it can affect the enzymatic and bacterial activity rates. Microbial growth is optimal under certain ranges of temperature, with important effects on biomineralization rates. A similar consideration is in place with the enzymes that may be denatured at temperatures higher than those in the optimal range. The pH affects the microbial growth and the enzymatic activity in a similar way as temperature. It is also crucial because of the ionic species that are going to be found in the medium needed for the formation of the mineral. The concentration of $Ca^{2+}$ is also a key exogenous factor, since it may be limiting the nucleation step, and it should be available in the near environment to the cell at the concentrations needed for the nucleation step. Additionally, it has been proved that higher concentrations of enzymes and/or microbial cells affect positively the amount of mineral formed [121,130–137].

On the other hand, BCM differs from BIM in the greater control over the biomineralization performed by the microorganisms [91]. The advantage of the use of this type of synthesis is that, as these biominerals are supposed to play an important role in the organism's survival and it is a process bound to high selective pressure, the final materials are generally very regular and present a high purity and high stability [11,91]. It is thought that the mineral structures obtained by the biological mechanism may have different properties than those obtained by the chemical synthesis [11,138]. If the genetic and metabolic signaling that allows this control over the mineral formation would be fully understood, it could be straightforward to obtain carbonates with desired properties from biotechnological approaches. For a full understanding of the control over the mineral properties, the importance of the secreted components and the cellular surface in the nucleation processes as a compliment to the metabolic pathways described has been studied.

## 4. Biopolymers Associated with Carbonates

### 4.1. Adsorption and Linkage of Metal

The EPS are generally made of high molecular weight molecules with charged functional groups and possess both adsorptive and adhesive functions. Thus, EPS can serve as a source of natural ligands that provide binding sites for other charged groups, including metals [139]. The EPS present a high ability for metal binding generally higher than mineral sorbents [140]. The ability and properties of microbial EPS have been studied and characterized [32]. The EPS and negatively loaded proteins in the cellular membranes can attract metal cations (e.g., $Ca^{2+}$, $Mg^{2+}$ and $Fe^{2+}$), increasing the ion concentration in the surrounding environment of the microbial cell. Such process may lead to ion supersaturation in the cell surrounding, which enhances the changes in the carbonate mineral nucleation and precipitation. However, as $Ca^{2+}$ can be attracted by electrostatic interactions, many other divalent metal cations could be attracted following the same physicochemical reasoning, such as $Cd^{2+}$, $Cu^{2+}$, $Fe^{2+}$, $Mg^{2+}$, $Mn^{2+}$, $Ni^{2+}$, $Pb^{2+}$ and $Sr^{2+}$. Once other metals are attached to the cell walls, or the proteinaceous S-layers and the conditions are favorable, such metal cations can slip in the previously formed crystalline structure following the ionic exchange

with Ca$^{2+}$. Such metal–mineral–microbe interactions can reshape the chemical nature of the carbonates though biomineralization to molecules other than calcium carbonates, such as strontium carbonates, magnesium carbonates or iron carbonates [7,39,65,67,141–148].

Many studies have concluded the main role of EPS in concentrating these divalent ions [7,67,69,145–147,149–153]. In an experiment that tried to characterize the differences in the interactions between biofilms and metal ions, when the biofilms (EPS) were deposited in (i) commercial plastics or (ii) periphyton (a natural substrate of EPS in freshwater environments), it was shown that EPS organic components, such as amines and phenolic compounds, could attract and bind metal ions of Cu(II) and Cd(II) very quickly. Additionally, amidic and aliphatic components of EPS showed a preferential binding to heavy metals [154]. The implications of the metal-binding properties of bacterial EPS and its possible role in the bioaccumulation of pollutants in the marine food chain were under study after the isolation and purification of EPS synthesized by species of *Marinobacter* species [104]. It was demonstrated such a type of EPS bind more efficiently Cu$^{2+}$ than Pb$^{2+}$ per mg of EPS, preferentially at a neutral pH. It was also shown that the adsorption increased with higher concentrations of the metals to bind, but certain factors such as increasing concentrations of NaCl decreased adsorption rates, as simplified in Figure 5.

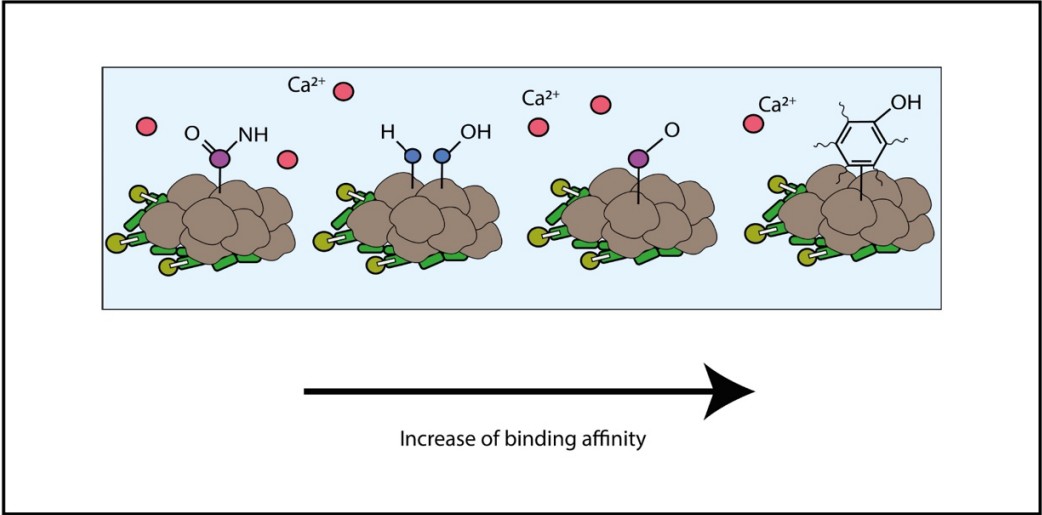

**Figure 5.** Simplified scheme showing the possibilities of interaction between charged molecules found in EPS and calcium ions, in this case. The red molecules represent any divalent ion, such as Ca$^{2+}$, Mg$^{2+}$, Fe$^{2+}$, Sr$^{2+}$, Mn$^{2+}$ or Ba$^{2+}$, found in the solution in the surroundings of the EPS–cells association. As the negative charges of the chemical groups present in the matrix increase their ionic strength, the binding affinity between the ions becomes stronger (from left to right).

Charged groups exposed on the cell walls of microorganisms, such as amino, carboxyl and phosphoryl groups, can have a role in the interactions with metal ions, especially at a neutral pH. In general, the amount of negative charges on the cell wall surfaces exceeds the number of positive charges, so the whole cell wall has an anionic nature [155]. This anionic superficial charge is responsible for the first stage of mineral formation. In order to balance the differences between negative charges on the cell walls and the positive charges of metals in the surrounding environment, which determine the decrease in free energy, the precipitation of metals occurs and leads to the formation of the mineral [156].

*4.2. Polymers Influencing Size, Morphology, Texture and Chemical Composition of Carbonate Minerals*

Since microbes are involved in biomineralization processes, much effort has been put into trying to understand how they can actually influence the final mineral formed. There is some evidence that superior organisms and microorganisms are able to tightly control biomineralization processes during MICP. Bioprecipitation starts with the formation

of nanoclusters and nanoparticle precursors. Different organisms such as mollusks can orchestrate the type of calcium carbonate polymorph that will result from the biomineralization. According to the free energy and enthalpy values of the different polymorphs of certain minerals such as carbonates, the differences are so small that reversions to the stability state can occur [157]. Taking this into account, a possible strategy followed by the microorganisms to control the final polymorph might be related to the control of the initial size of the crystals [157]. This step could be biotechnologically manipulated by monitoring the ionic strength, supplying controlled concentrations of reactants, and by adding certain organic and inorganic molecules to the growth medium. Therefore, obtaining the desired polymorph with relatively uniform size distributions could be feasible. Calcium carbonate polymorphs constitute a special case: vaterite is less dense than calcite, and aragonite is denser than calcite. Vaterite and aragonite are formed as metastable phases in the biomineralization processes, being the most stable polymorph calcite. The addition of certain impurities, such as $Mg^{2+}$, to the mineral seed surfaces could lead to changes in the superficial energy and finally influence the polymorph formed. Considering that the three main polymorphs of calcium carbonate minerals have very similar free energy, the addition of certain impurities can hamper their stability and control the kinetic and thermodynamics of the crystallization and which phase crystallizes in the end [157]. It has also been demonstrated that alcohols can affect the crystal morphologies, polymorphs [158,159] and Mg composition [160].

The presence of microorganisms may deeply influence the mineral crystal size, morphology and type of polymorph. Liu et al. (2021) [100] obtained and characterized crystals of amorphous calcium carbonate (ACC) and vaterite carbonate polymorphs synthesized by *Bacillus subtilis* in the presence of organic matter, nanometric crystals of different shapes and elemental compositions nucleated in an organic matrix. The influence of the $Ca^{2+}$ concentration positively affected the biomineralization rate, and a $Ca^{2+}$ concentration above 0.8% affected the quantity, morphology and structure of $CaCO_3$ crystals. The EPS might potentiate the formation of vaterite, regarding the formation of inorganic–organic complexes. On the contrary, proteins released in the medium resulted in inhibition of the stabilization of the unstable/metastable polymorphs into calcite. Braissant et al. (2003) [161] demonstrated the role of EPS and L-amino acids present in the medium determining the size and shape of minerals precipitated by *Xanthobacter autotrophicus*. Changes in the concentrations and composition of EPS lead to precipitated calcite and vaterite crystals with different spatial configurations. Chekroun et al. (2004) [127] showed that the presence of calcite and vaterite crystals with different morphologies were associated to the presence of living or dead cells and that the organic molecules promoted calcium carbonate growth. Coccolithophores and their ability to create small crystals of calcite (coccoliths) in marine habitats have also been related to MICP processes. In the model species *Pleurochrysis carterae*, a unicellular microalgae, it was shown that the interaction of the coccoliths with vertical gel-like structures inside the Golgi contributed to the final morphology of the mineral [162]. This interaction was combined with the 3D structural action of the coccolith lipidic membrane.

Specific macromolecules can control the crystal phase or even shape the lattice structure that forms the crystal texture, size and orientation of single-crystal domains and the macroscopic shape of the crystals [23,163–165]. The influence of certain biopolymers in the crystallization of calcium carbonates was tested [166], determining that the influence of nongelling biopolymers (xantan and gellan) and stack-line rhombohedra aggregates were formed, while in the presence of gelling biopolymers (pectin, sodium alginate and κ-carrageenan) influenced rosette-like aggregates. The concentration of these biopolymers also affected the propensity to nucleation. Additionally, microbial EPS composed of acid polysaccharides has been demonstrated as influential in the nucleation and precipitation of amorphous phases (ACC) that precede the formation of more thermodynamically stable phases (e.g., calcite or high Mg-calcite) [167].

Eukaryotic lysozyme can play an important role in calcification in avian eggshells [168]. Experimental approaches in determining the influence of this enzyme in the morphology

and size of carbonate crystals [168] demonstrating that, under non-biotic conditions, hexagonal platelets of vaterite and spherical vaterite aggregates were the main components of the precipitate, being crystal faces well expressed in the hexagonal direction. In contrast, in the presence of lysozyme, calcite was favored to precipitate after dissolution-crystallization starting from vaterite, in the same conditions and after the same time reaction, and this effect was dependent on lysozyme concentration. Thus, lysozyme might accelerate the precipitation process and influence the final polymorph.

Hernández-Hernández et al. (2008) [169], investigating the effect of globular proteins, such as lysozyme, myoglobin, alpha-lactalbumin and ribonuclease-A, on $CaCO_3$ precipitation, found that the superficial charge and type and quantity of ionizable charged amino acids enhanced the biomineralization process and proposed the following reasons for such an effect: (i) the presence of different charges on the protein surface led to different preferential adsorption of these proteins to specific crystal faces, decreasing the mineral formation rate; (ii) the effects of the morphology of the calcite growth varied depending on the nature of the charges; (iii) all the proteins favored precipitation of calcite among the different types of polymorphs; (iv) acidic proteins at low concentrations inhibited the precipitation of vaterite and calcite. It was proposed that acidic proteins tend to act as nucleation sites for calcite because of the strong attraction of $Ca^{2+}$ ions to the surface. However, in the case of alpha-lactalbumin, its high affinity acted as a chelate, sequestering the calcium ions and, thus, reducing the precipitation. Interestingly, lysozyme, resulting from such a study, enhanced the precipitation rate even though it was positively charged.

In some recent studies, the control of MICP by ureolytic-driven activities has been investigated [170,171]. The expression of a green fluorescent protein (GFP) was used as a genetic marker of MICP to compare the biomineralization performances of *Sporosarcina pasteurii*, with low vs. high ureolytic activity rates [170]. In another recent work [171], it was demonstrated for the first time that the morphology and nanomechanical properties of $CaCO_3$ can be adapted if the metabolic activities and precipitation kinetics of certain ureolytic microorganisms are modulated or controlled via genetic engineering. It was hypothesized that microorganisms with low urease activity rates would produce larger calcite crystals. Two genetically engineered strains of *E. coli* were compared with natural *S. pasteurii* strains, well known for their MICP capacity. The results highlighted that the modified strains were able to precipitate the calcite crystals, firstly performing a quick and almost complete depletion of calcium ions present in the medium and, at the same time, an increased average size of the crystal. This higher concentration of calcite did not affect the viability of the strains.

These initial studies demonstrated that the interactions of microorganisms and organic compounds, including EPS and enzymes, are important for defining the rates and properties of MICP and provided the proof of concept that the modulation of the final characteristics of the precipitated minerals is possible by manipulating microorganisms even by genetic modifications.

### 4.3. Polymeric Substances Secreted by Microorganisms and Micritization, Lithification and Porosity Processes

Microbial geoactive interaction with the environment has been widely studied for understanding the biotransformation of elements, biogeochemical cycling and metal and mineral transformations [172,173] as well as from an application perspective for enhancing and controlling biomining or bioleaching processes [39]. Because of the variety of properties displayed on their charged surfaces and their versatile metabolic activities, microorganisms can induce profound changes in the metal composition in their environment, or they can enhance or mitigate the toxicity of such metals in the environment as well as modulate mineral precipitation and formation, dissolution or deterioration [174,175]. Metals and/or minerals are directly involved as passive and/or active promoters of the microbial metabolism [39], also in relation to the metal speciation, the changes in which can cause an increase or decrease in the metal mobility [176]. Metal speciation is directly influenced by

microorganisms that use different processes to modulate the chemical state of the minerals, including direct redox transformations, protein–ligand binding reactions, organic- and inorganic-mediated precipitation, active transport, externalization and intracellular compartmentalization. The metal-binding abilities that present the negatively charged proteins are particularly enriched at the cell wall level and especially at membrane levels, either on the external cellular membrane of those delimiting organelles or cell compartments inside the cell [44,45].

Among the different types of minerals, carbonates are strongly influenced by microorganisms for their turnover, precipitation and solubilization. A quite large range of microorganisms, including eukaryotic algae, fungi and bacteria, can have important roles in the precipitation of carbonates, such as aragonite ($CaCO_3$), dolomite [Ca, $Mg(CO_3)_2$], calcite ($CaCO_3$) and Mg-calcite as well as vaterite ($CaCO_3$) and ACC [65,67,152]. Once carbonate precipitates are formed, several geochemical and geological processes modify the initial crystalline structure in order to stabilize the mineral at the nano- and micrometric-scale, in a process known as 'micritization'.

Micritization, known to occur in marine shallow waters, can be defined as a diagenetic process in which the original fabric of carbonate grains is gradually altered to cryptocrystalline textures by the repeated cyclical boring and filling of boreholes with micritic precipitates catalyzed by the interaction of abiotic and microbially mediated processes [177,178]. It was thought that biotic processes at the nanometric and micrometric scale follow the same path as the abiotic process according to nucleation theory; the formation of the mineral occurs by the attraction of ions to affine molecules in a supersaturated solution, and the consequent formation of chemical boundaries initiate the physical shaping of a nucleus, also with pH and temperature optimal conditions [178]. Due to the fact of electrostatic interactions, the initial nucleus would start attracting more and more ions from the solution, and the precipitation and growth of crystals would eventually occur [179]. The selection of specific isomorphs is determined by the superficial free energy followed by the orientation of the initial nucleus in space. The following interactions between mineral particles and the variable physicochemical changes in the near environment would further modify the metastable isomorphs in order to reach the most stable mineral phase to be stabilized at the minimum free energy (e.g., in case of carbonates, from vaterite, aragonite or low Mg-calcite to calcite or high Mg-calcite). Nonetheless, several studies have suggested that such a process is a simplified view of what is indeed occurring during biomineralization and that additional more complex steps are needed for achieving the final crystal structure. For example, it has been observed in the biotic processes of teeth formation in some organisms that an amorphous carbonate phase, such as ACC, first occurs as a precursor to the final mineral, not necessarily through a nucleation process [180–182]. In such a type of mineral formation pathway, the initial nucleus leads to an amorphous momentaneous stable phase. Geological records found in the Wolfenden tufa (British Columbia) suggest that the precipitation of nanocrystalline calcite was preceded by the precipitation of ACC and an intermediate monohydrocalcite (MHC). This finding was astonishing, since ACC, as a first stage to nucleation and precipitation of calcite, is not usually observed in nature. The ACC effect in nucleation and precipitation of aragonite has been mostly described in cold water environments [183]. This record also provides insight into the nucleation and depositional effects of microbial biofilm (EPS), resulting from the bacterial metabolism, that provided nucleation sites because of the acid polysaccharides found in the matrix may stabilize the formation of the ACC [167]. Although amorphous precursors have been considered and studied in biomineralization processes, the formation of an ionic nucleus cannot be excluded from the process of mineral synthesis. Despite analogies with the abiotic mineralization process, minerals resulting from biotically mediated processes exhibit higher complexity because of the many different interactions with the organic matrix and biological compounds, which influence the final size, shape, polymorph or physicochemical properties of the minerals that, as a whole, can have rather different properties from those produced abiotically [184].

Endolithic microorganisms capable of boring minerals have been associated to the micritization, lithification and coastal erosion processes in marine environments [185]. Such endoliths encompass a large taxonomically and evolutionary nonhomogeneous group of microorganisms, including photosynthetic cyanophytes, eukaryotic green/red algae and heterotrophic fungi, that actively penetrate carbonate substrates and actively rebuild the minerals. These groups of endolithic microorganisms are widespread in different geological areas throughout the Phanerozoic record [185]. A mechanism proposed for the precipitation of calcium carbonates is mediated by P-type ATPases, which modulate $Ca^{2+}$ concentrations by exchanging these ions against a concentration gradient for obtaining energy [186]. The ultimate reason(s) to perform such boring activity is debated, and the acquisition of nutrients, the competition for resources, the protection from physically extreme conditions such as UV radiation, escaping from predation or the prevention of detachment from rocks because of water currents have been proposed, among others [187]. For instance, in microbial mats it has been shown that EPS remains attached to the $CaCO_3$ formed as a consequence of the microbial metabolism, and, thus, it could be expected that its presence might influence the late diagenetic processes [62,188–191].

Another relevant example of the contribution of microorganisms to porosity, lithification and micritization processes is found in microbialites. Microbialites are extremely diverse benthic sedimentary deposits made of carbonate mud-sized crystals (with a particle diameter of less than 5 μm) formed by the mediation of microorganisms [192]. The diversity of microbialites, which can vary depending on the type of paleoenvironment, can be explained by the enormous heterogeneity of microorganisms forming the ancient microbial mats or microbial communities where these sediments were formed. Thus, the "trapping and binding" activities mediated by the microbial cell walls and EPS, which can interact with metals and ions and provide the physicochemical conditions to determine the mineral precipitation, vary with the microbial types and the occurring environment and led to the formation of different types of microbialites, such as oncoids, thrombolites, skeletal stromatolites, agglutinated stromatolites, micritic-evaporitic stromatolites and fenestral laminites [193,194]. For instance, in the case of the oncoids, the most common microbialites and the conditions for their formation in most freshwater habitats are related to the high saturation of carbonate ions, which combined with the metabolic activity of cyanobacteria led to the precipitation of the calcium carbonates [195]. Nonetheless, the grainy faces observed in this type of microbialites, combined with the continuous laminae and truncations in the oncoid cortices, suggest that they accumulated in waved or hectic environments, but the appearance of discontinuous laminae and stromatolitic overgrowths might indicate that, in some periods, the hydrodynamic forces were not enough to overturn the oncoids [196,197]. Such observations highlight the influence of environmental conditions on the formation of microbialites.

Reid et al. (2000) [198] defined a series of mechanisms implicated in the formation of modern and possibly ancient stromatolites by an interplay between the stages of sedimentation and periods of lithification associated with a succession of microbial communities. The lithification stages were suggested to be dependent on two main microbial processes: (i) heterotrophic bacterial respiration and (ii) photosynthetic activity by present cyanobacteria. Thus, the laminated structures observed in stromatolites could be related to a micritization process in surface biofilms occurring during discontinuities in sedimentation processes [198]. The same authors proposed that microboring and precipitation are related to the presence of thicker layers of fused grains that might result from the degradation of the polymer in boreholes. Such observations provide new insight into the role of microbes and lithification for the formation of the stromatolites in the Precambrian [199].

Microbial mats trap and bind particles, as well they induce mineral precipitation, both processes that contribute to the lithification of the mat. When mats are lithified, they start being known as microbialites or stromatolites, which are fossilized [200]. This is why microbial mats are considered living analogues and potential precursors of stromatolites, which were among the first biological structures capable of MICP established on the ancient

Earth [201]. By reconstructing the chemical metabolic signatures performed by different microorganisms fossilized in these stromatolites, it would be possible to learn more about how the ancient atmosphere or ancient seas were composed. Moreover, stromatolites contain the geochemical key to our understanding of the evolution of microbial life on Earth [153]. As a remarkable example, the characterization of the cyanobacteria *Geitlerinema* sp., as one of the dominant microorganisms found in microbial mats and analogous stromatolites, has led to the suggestion that these microorganisms greatly contributed to the oxygenation of the ancient Earth's atmosphere [153]. Before this event took place, some hints in fossilized stromatolites have proved that arsenic was a chemical choice for ancient photosynthesizers and respirators. Experiments and measures of a purple microbial mat in Laguna la Brava, in the Atacama Desert in Chile, showed high levels of arsenic, sulfur and lithium but not oxygen [202]. The experiments performed with microorganisms isolated from the microbial mat led to the conclusion of the existence of an arsenic cycle linking photosynthesis and respiration carried out by purple sulfur bacteria and arsenate-reducing bacteria in what is called arsenic-cycling mats. Since the Atacama Desert shares so many similarities with extraterrestrial bodies such as Mars, the understanding of current and ancient life in this ecosystem might shed light onto life on other planets [203]. The approaches used on living microbial mats can be adapted for understanding more the composition of ancient microbial mats. These methods mainly consist of the analysis of fossilized proteins embedded in ancient carbonates [204–206] and even polysaccharides well preserved into the mineral structures [207], hints that could lead to a better comprehension of the formation processes of modern and ancient microbialites. Further research on the taxonomy of microorganisms involved in the diagenetic processes will lead to a better understanding of the role of organics secreted by microorganisms and how their properties and chemistry may influence mineral precipitation, growth and evolving changes in carbonate structure that are observed in many rock samples from different geographical locations around the globe. Not limited to that, these studies may increase our knowledge regarding our ancient past.

## 5. Implications and Applications

### 5.1. Geology: Early Diagenesis, Burial Diagenesis, Carbonate Reservoirs and Ancient Carbonates

5.1.1. Diagenesis

Microorganisms drive very many geochemical processes thereby influencing the geological processes. As presented above, microorganisms are implicated in mineral growth, crystallization, nucleation and dissolution [16]. They transform organic carbon in the sediments, contributing to the formation of fossil hydrocarbon deposits, and they cycle the elements in the ocean and on land and weather rock and minerals [17]. To perform such processes, microorganisms are organized into complex communities that may have a precise hierarchical organization where the different microbial groups that coordinate in a metabolic network reside. For instance, the already introduced microbial mats are complex assemblages of microorganisms mostly stratified into layers with a discrete thickness. They are usually composed of oxygenic photosynthesizers, aerobic heterotrophs, anoxygenic phototrophs and anaerobic heterotrophs [88]. The metabolic network and the flows of nutrients and metabolites are very complex and deeply influence mineralization, the formation of carbonate sediments and the early diagenetic processes. Even though bacteria can induce the precipitation of carbonate by binding calcium ions and form mineral structures, their influence on the sedimentary process needs to be elucidated. Piggot, A. (2014) studied the bacterial communities in certain carbonate depositional environments to find any relationship with the early diagenesis. By means of different laboratory-scale experiments, it was proved that early diagenetic processes in hypersaline lagoons are the result of bacterial sulfate reduction activity and the degradation of EPS by heterotrophic bacteria [208]. Environmental factors, such as salinity and sediment bioturbation by animals, such as crabs or other crustaceans, or other organisms, could also influence the conditions of diagenesis. For instance, diatoms (phytoplankton) are usually embedded in the first layers of microbial mats, creating interfaces between the lithified mats

and silicon, highlighting a relationship between EPS and the transit to a biomineralized phase associated to Mg-Si minerals. Diatoms would be the main source of Si, and the EPS and organic matter would act as Mg concentrators [200].

One of the many diagenetic processes in which microorganisms are thought to be rather involved is burial diagenesis. It begins when sedimentary sequences are buried beneath the reach of surface-related processes. The investigations of burial diagenesis have been developed for hydrocarbon exploration through a series of steps (detailed in [209]), which include the characterization of parameters such as porosity and permeability distribution, petrographic and geochemical analysis, as well as research regarding burial history and its modeling.

The analysis of microorganisms and microbial communities can complement the study of burial diagenetic processes, being microorganisms' indicators of specific geochemical processes [209]. Micritized carbonate grains formed in a second burial diagenetic process in shallow marine environments inhabited by the living organisms *Halimeda incrassata* and *Amaeana Angulus* can be a result of the recrystallization of the aragonite skeletal rods and needles, derived from the skeletons of dead animals, into low Mg-calcite composite of pseudo-micrite, mini-micrite and micrite [210]. The textural and mineralogical changes to the skeletal carbonates are contributed by chemical reactions of organic materials and microboring activity. The degradation of the organic matter of the dead animals by oxidative or reductive microbial metabolic processes, such as oxygenic respiration, sulfide oxidation or sulfate reduction, can promote carbonate dissolution, precipitation and recrystallization [210]. The extent of these processes varies according to the specific geographical area in the function of the size of the sediments, as "the more muddy the sediment is, the extent of cryptocrystalline carbonate along the sand grain fractions is minor" [210].

Mineral boring is widely contributed to by the activity of microorganisms, which may produce characteristic boring patterns depending on the specific taxa. Perforations of the mineral matrices can be produced by acidic excretion via the microbial cells and by sequestration of calcium ions via calcium pumps located on the cell membranes [186,187]. Boring microorganisms are able to adopt longitudinal/horizontal dispositions in different sediments. The spatial distributions of boring microorganisms in the sediments are driven by several factors, including the region, the latitude and the depth of the specific sediment layers; thus, the boring patterns are indicators of the geographical, climatic and bathymetrical origin of the sediments. Environmental factors, such as light intensity and water supply, among many others, influence the boring activity of microorganisms, including the emptying of the bore hole, secondary carbonate precipitations, secondary leaching within the boring, diagenetic alteration of host substrates (replacement, dolomitization and silicification), and the infilling of microborings by phosphatized chalk or limonite to form natural casts [185,187]. Boring microorganisms are suggested to participate in the recrystallization processes of the skeletal shells in a combination with the action of microboring cyanobacteria and the formation of a mucilaginous cover formed by the organisms that create a microenvironment in which the microorganisms and gasses become trapped [210]. Secondary diagenetic process may not increase the stability of the carbonate rock in formation, since it translates a metastable carbonate isomorph (e.g., aragonite) to another metastable isomorph (e.g., low Mg-calcite), in contrast with what is usually expected for rock consolidation. Microorganisms leave perforations, grooves and tunnels within hard carbonate substrates, and due to their very small size, they are very easily fossilized. For example, it has been suggested that algal borings could be very useful in the definition of ancient photic zones [211]. With that purpose, Swinchatt (1969) [212] supported in his work that the values of different parameters in microborings could be paleobathymetry indicators of water depths of less than 15–20 m. Nevertheless, not only phosynthesizer microorganisms would be considered as microborings, since micritization in the absence of light, performed by heterotrophs, chemotrophs and fungi, may likely happen as well.

Endolithic microorganisms can also be easily confused with ambient inclusion trails (AITs), usually found in Precambrian rocks but also in younger records. AITs are mi-

crotubular structures formed by the migration of the mineral crystal through a lithified substrate, in this case carbonate rocks. Even though AITs appeared after endolith microborings in the time scale, AITs are morphologically very similar to endolithic fossils or microborings, and so their appearance might interfere in the geological interpretations of rock samples. Additionally, they have not been extensively studied, and it is not yet well known if only chemical processes or biological processes are involved in their formation [213]. In a recent study [214], bioturbation records of an endolithic nature (related to *Endochonchia lata*) and AITs were found simultaneously, co-occurring in small skeletal shells. This co-occurrence allowed the determination of some morphological, chronological and distributional differences between these two kinds of sediment perforations.

5.1.2. Carbonate Reservoirs

The demand for oil and fossil fuels is persisting, with the need for more ecofriendly and efficient methods for oil recovery. Oil and gas exist as fluids in the pore's spaces of subsurface sedimentary rocks, e.g., sandstone and carbonate rocks. The current methods recover just 45% of the original oil in place (OOIP), with 55% remaining in the reservoirs. Therefore, tertiary strategies in which thermal, chemical and/or microbial processes are applied to increase the recovery percentage are needed [215]. The use of microbially enhanced oil recovery (MEOR) techniques have been proved as an efficient, useful and attractive alternative strategy. MEOR is based on two main principles: (i) enhancing the oil movement throughout the porous media by altering the interfacial properties of the subsurface by decreasing interfacial tensions, increasing the reservoir pressure, increasing the fluidity of the mobility of those fluids; and (ii) by upgrading the oil quality by degradation and removal of heavy metals and sulfur on heavy oils. Such principles can be implemented by exploiting the functional capacities of microorganisms. MEOR is usually performed via the injection of certain microorganisms inside the pores of reservoirs. Microorganisms are vehiculated in an appropriate growth medium with the required nutrients. Such an approach can lead to a recovery of up to 50% of the residual oil remaining in the reservoir after the primary and secondary extraction treatments [216]. Carbonate reservoirs contain half of the world's petroleum reserves and MEOR may provide an important contribution to recovering such reserves [217]. However, few studies on MEOR techniques for oil recovery have been implemented until now because of the complex pore structures and hydrodynamics that exist in most reservoirs. The main differences between carbonate reservoirs and sandstone reservoirs are the chemical properties and porosity configurations. These configurations depend on primary (depending on the degree of initial cementation and compaction) and secondary (related to recrystallization, dissolution, dolomitization and fracturing, among others) factors [218].

When the conditions of the reservoir pores are favorable for the injection of microorganisms, the selected microorganisms can be regulated for the production of different metabolic products, with different effects on the oil recovery. The regulation of the microorganism's metabolism depends on properties such as the porosity of the reservoir, permeability, pH, pressure, temperature and the nutrients present in the pore. MEOR is included in the nonthermal subgroup of EOR processes, being one of the most efficient thanks to an extraction procedure enhanced by the chemical and physical changes in the rock pores and the composition of the residual oil induced by the microbial growth [219]. Zobell et al. (1964) [220] resumed the influence of different microbial products in these processes:

1. Dissolution of inorganic carbonates by various microbial metabolites.
2. Generation of bacterial gases, influencing a decrease in the oil's viscosity and, therefore, enhancing its flow through the pores.
3. Production of surface-active compounds.
4. High affinity of bacteria for solids, so the bacteria are able to replace the oil attached to the rock surface and to scroll the oil to the center of the pore to facilitate its flow and removal.

All of these microbial-mediated processes lead to a classification of the most important MEOR mechanisms in (i) the changes in the oil–rock–water interfacial properties related to the reduction of the interfacial tension (IFT) and wettability alteration and (ii) changes in the flow behavior by the plugging of high wettability zones, increase in the wettability by the production of organic acids, and the production of gas and microbial solvents in situ [221]. In the specific case of carbonate reservoirs, microbial acid production and microbial gas production have been proposed also suited for MEOR processes [218]. The oil recovery is highly dependent on the rock pore size. In heterogeneous oil reservoirs, two steps have been proposed for the enhancement of oil recovery, which involve MEOR and chemically enhanced oil recovery (cEOR): in a first stage, the oil trapped in cm/mm scale pores might be removed thanks to the use of biosurfactants produced by the microbial metabolism's stimulation, and when oil is no longer recovered; as a second stage, alkali or chemical surfactants are introduced on narrow pores (micrometric scale) [222–225].

Many microorganisms, including mixed cultures of bacteria and archaea, can be implemented depending on the metabolic products desired to improve the oil recovery, usually consisting of facultative anaerobic bacteria mixed cultures [170]. Nevertheless, the potential of the implementation of certain types of microorganisms, such as thermotolerant [226], halotolerant [227] and ultra microbacteria [228], has also been suggested because of the need for strains with high tolerance to survive to the extreme ambient conditions, reduced size to avoid plugging problems, and high metabolic activity for high recovery rates. *Clostridium* is one of the more used genera due to the fact of its ability to produce spores resistant to the extreme conditions of the reservoir pores [229]. *Bacillus* genus is also an option because of its skill to produce biosurfactants, which are very useful metabolic products that are able to enhance the displacement of oil by three fundamental mechanisms: (i) reduction of the oil–rock and oil–brine IFTs; (ii) changes in the rock wettability; and (iii) emulsification of the crude oil [230,231]. Biosurfactants are so important that many strategies for their production in the core of the reservoirs have been designed. One of them is direct injection inside the reservoir pores synthesized at an industrial scale [232,233], which is still trying to be optimized to reduce the costs associated with the bio-reactor maintenance and purification of the final product, but it nevertheless has very good results in laboratory approaches. A more efficient strategy is the injection of biosurfactant-producing microorganisms in the reservoir [216,234]. This is possible if the temperature and pressure conditions of the reservoir are compatible with the microorganisms used and, additionally, it should be accompanied by the injection of nutrients [235,236]. These two last strategies are considered as efficient when using the indigenous microorganisms found in the rock because of their extremophilic natures. Among the different types of biosurfactants that can be used, lipopeptide-based biosurfactants have been proven experimentally to be very effective compared to other biosurfactants, such as rhamnolipids [237].

The advantages of the application of microorganisms and their surfactants are the attractiveness for application in oil and already-exploited reservoirs, the lack of need for new infrastructures, low requirements of energy, the possibility to optimize the applications for carbonate reservoirs, the benefits amplified with time (the opposite for cEOR techniques), and the environmentally friendly nature. Nonetheless, more research is needed in order to optimize the process and avoid current inconveniences, such as the high dependence on the physical and chemical properties of each reservoir, its low speed and its uncertain success at extreme conditions of the reservoir [218,238].

*5.2. Engineering: Bioremediation, Plastics and Biomining*

5.2.1. Bioremediation by Carbon Dioxide Fixation

An important biotechnological application of biogenic carbonate formation is related to bioremediation processes in relation to the growing levels of $CO_2$ in the Earth's atmosphere as a consequence of the use of nonrenewable resources (i.e., oil and gases) to obtain energy [239]. Carbon capture and storage (CCS) technologies for the reduction of the existing $CO_2$ in the environment have the potential to promote a reduction in greenhouse gas

emissions [240]. $CO_2$ sequestration consists of the fixation of $CO_2$ in geological formations and finally storing it by injecting the gas into deep coral beds, saline aquifers and oil/gas reservoirs [240]. The most thermodynamically stable minerals for $CO_2$ storage are calcium and/or magnesium carbonates, which can allow long-term stability [241]. Two strategies can be followed: (i) in situ belowground mineralization consists of directly injecting $CO_2$ in gas or aqueous solution into these geological reservoirs so it reacts naturally with alkaline metals (e.g., $Ca^{2+}$ and $Mg^{2+}$) to form the carbonate minerals; (ii) ex situ aboveground mineralization which implies a series of procedures where the alkaline metals are extracted, usually from natural silicates or industrial/waste by-products, and allowed to react with aqueous $CO_2$ [241]. The Reactions (8)–(12) are involved in this chemical process:

$$CO_2 \text{ (g)} \Leftrightarrow CO_2 \text{ (aq)} \tag{8}$$

$$CO_2 \text{ (aq)} + H_2O \Leftrightarrow H_2CO_3 \tag{9}$$

$$H_2CO_3 \Leftrightarrow H^+ + HCO_3^- \tag{10}$$

$$HCO_3^- \Leftrightarrow H^+ + CO_3^{2-} \tag{11}$$

$$CO_3^{2-} + Ca^{2+} \rightarrow CaCO_3 \text{ (precipitates)} \tag{12}$$

Following the present schematic Reaction (13):

$$\text{Metal oxide} + CO_2 \rightarrow \text{Mineral carbonate} + \text{Heat} \tag{13}$$

In situ mineralization presents the problem of $CO_2$ leakage, a very common inconvenience and, therefore, it implies special attention to the quantity of $CO_2$ fixed and continuous monitoring of the fixed carbon. Unlike in situ mineralization, ex situ mineralization offers an alternative with many advantages, since alkaline metals can be obtained from industrial by-products or waste products, such as stainless-steel fittings, coal fly ashes and cement and lime kiln dust. In turn, two routes are considered for ex situ mineralization: (i) direct, the simplest carbonation method, enhanced mechanically and thermally, in which Ca or Mg react with $CO_2$ in a single step; (ii) indirect, taking two or more steps for the carbonation due to the initial extraction of the alkali minerals, followed by the precipitation reaction between $Ca^{2+}$ or $Mg^{2+}$ and $CO_2$ (aq), providing high-purity carbonates. The advantages of this approach are the reuse and recycling of industrial wastes known for their harmful effects to the environment, which helps the planet, the costs optimization of processes that requires great amounts of energy for mining and extracting the minerals, and the recycling of additives and catalysts. Additionally, the final product (i.e., $CaCO_3$ or $(Mg,Ca)CO_3$) is a very highly valued product used in industries such as paper or monument/heritage conservation, being very versatile depending on the carbonate mineral size, color, shape, polymorph, brightness and density. Obtaining the desired carbonate mineral with specific properties can be achieved by controlling parameters in the synthesis processes, such as temperature, pH and the presence of impurities, or by controlling the crystallization step using ultrasounds. The use of certain microorganisms to obtain specific carbonate minerals and polymorph(s) can also be considered [242].

Studies with isolated proteins as fixation catalysts for EICP have also been tested [243,244]. One recurrent enzyme is carbonic anhydrase (CA), and ubiquitous zinc-dependent metalloenzyme with lipase activity, which is produced by many (micro)organisms. This enzyme has roles in multiple cellular functions and metabolic pathways, such as photosynthesis, respiration, $CO_2$ -$HCO_3^-$ homeostasis in blood and tissues, as well as the transport of $CO_2$ to the rest of the tissues in higher organisms [245]. $CO_2$ sequestration by microbial CA has been proved in various studies [246–249]. The CA catalyzes Reaction (9), which is the slowest and, thus, the limiting step:

$$\text{CA-ZnH}_2O \Leftrightarrow \text{CAZnOH}^- + H^+ \tag{14}$$

$$\text{CA-ZnOH}^- + \text{CO}_2 \Leftrightarrow \text{CA-ZnHCO}_3^- \tag{15}$$

$$\text{CA-ZnHCO}_3^- + \text{H}_2\text{O} \Leftrightarrow \text{CA-ZnH}_2\text{O} + \text{HCO}_3^- \tag{16}$$

As it can be observed, Reaction (9) is substituted by Reactions (14)–(16) and, therefore, the reaction speed increases significantly. CA not only enhances the hydration of $CO_2$ but boosts $CaCO_3$ precipitation [248].

The main bulk of the carbonate pool on Earth is of biotic origin [250], and even though some habitats are oversaturated with $HCO_3^-$ and $Ca^{2+}$ ions, carbonate mineral formation does not take place in the absence of certain microorganisms, or the biotic production affects differently the final carbonate formed [251]. As seen before, many microorganisms can induce carbonate precipitation, such as autotrophic (cyanobacteria), aerobic heterotrophic (*Vibrio*, *Bacillus* and *Pseudomonas*) and anaerobic heterotrophic (e.g., sulfate reducing bacteria and methanogenic archaea). MICP by calcifying cyanobacteria can have implications on $CO_2$ sequestration [252]. Cyanobacteria are photosynthetic Gram-bacteria that can be found in numerous environments, from terrestrial to marine, even in the most extreme environments, such as deserts, frozen soils (i.e., permafrost) or deep-sea sediments [106]. Cyanobacteria are responsible for half of the global photosynthetic processes [253,254] in the form of phytoplankton in the sea, and they play a key role in the carbon cycle, and according to the oldest fossil records available, they are candidates responsible for the current redox conditions of the Earth's atmosphere. The special characteristic of cyanobacteria and eukaryotic algae is that they present a carbon concentrating mechanism (CCM), a metabolic system that allows the cells to enrich $CO_2$ in the small environments inside and outside of the cells [253,255], favoring the interaction of the gas with the enzyme Rubisco for its fixation into organic carbon structures [255]. The carbon capture pathways depend on gas diffusion through the action of different membrane transporters [256]. By keeping a constant flux of $CO_2$–$HCO_3^-$ and hydration of these two species, due to the action of CA, a supersaturation of $HCO_3^-$ is maintained.

The $Ca^{2+}$ supersaturation index is also a key factor. Calcium is a very important second messenger in many cellular processes, so it is crucial for cells to be able to regulate its internal and external concentrations. In addition to the supersaturation, a high pH in the immediate external environment is also important for the biomineralization process. The pH conditions that enhance the precipitation of $CaCO_3$ are created within the EPS and the S-layer by the action of CA, which consumes $H^+$ and produces $OH^-$ groups, the latter being secreted to the medium. In addition, the photosynthetic electron transport (PET) and the $H^+/Ca^{2+}$ antiport transport system determine the conditions for MICP [254,257]. Thus, the calcification process depends on the (i) photosynthetic activity, (ii) CCM, (iii) extracellular surface properties and (iv) general environmental conditions.

The use of calcifying cyanobacteria provides certain advantages with regard to other microorganisms:

1.  Cyanobacteria and microalgae assimilate large quantities of $CO_2$ and tolerate high $CO_2$ concentrations, being able to use $CO_2$ sources such as fuel gas.
2.  Many species are unaffected by products such as the $NO_x$ and $SO_x$ present in industrial carbon sources such as flue gas, so they can be used in the treatment of that kind of product easily and in a cheap way.
3.  Halophilic cyanobacteria can be cultured in seawater, thus saving freshwater.
4.  Thermophilic cyanobacteria operate at high temperatures, saving fuel gas cooling.
5.  They can be genetically manipulated relatively easily for performance improvement.
6.  Nutrients for bacterial growth can be supplied through recycled wastewater.

McCutcheon et al. (2019), in laboratory-scale experiments, induced precipitation of magnesite and other magnesium carbonates for $CO_2$ storage from wastewaters [258]. This study provides the first proof of the possibility of using MICP via magnesium carbonate precipitation for $CO_2$ sequestration, achieving a storage of 238 tons of $CO_2$ per hectare of wetland per year. As the crystals were coated by EPS, their importance in the nucleation

process was demonstrated. Several other studies followed to set the conditions for the process implementation at an industrial scale [184,259–262].

Nevertheless, further research is needed for the process's optimization, especially on the biochemical and molecular mechanisms that regulate the calcification process, before scaling the process to an industrial level.

### 5.2.2. Bioremediation by Removal of Heavy Metals from Contaminated Soils

Due to the fact of urbanization and excessive anthropogenic activity, large quantities of heavy metals are very often found in soils and water, which need to be constantly monitored in order to detect and remove those hazardous contaminants. Another big contribution of carbonates to bioremediation is the removal of heavy metals and radionuclides from contaminated habitats, such as soils or underground water. In the past few decades, several strategies for the removal of contaminating metals in soils have been developed, with solidification–stabilization, electrokinetic extraction, soil flushing and phytoremediation the most studied. Still, these methods are extremely expensive, needing large inputs of chemical substrates and energy, and rather inefficient. MICP is in consideration as an alternative to these processes. During the MICP, heavy metals such as divalent ions are substituted by calcium in the crystalline structure of the carbonate precipitate, changing the bioavailability of these metals from bioavailable to a nonbioavailable state. In the removal of contaminating metals, CA, alkaline phosphatase, urease and asparaginase have been tested, with urease being the most cost-efficient enzymatic biosystem [130,263–268]. *Sporosarcina pasteurii* is the workhouse in most of the studies because of its high urease-activity rates [115,263]. In the case of underground water, the divalent ions of heavy metals, such as $Sr^{2+}$, $Cd^{2+}$, $Pb^{2+}$, $Mn^{2+}$, $Zn^{2+}$ or $Fe^{2+}$, would competitively coprecipitate in the formation of $CaCO_3$ minerals with the addition of these divalent ions to the calcite lattice. Thus, the formed heavy metal-carbonate precipitates can be easily recovered mechanically from the water. Kang et al. (2014) [269] studied 55 calcite-forming bacteria from an abandoned mine under laboratory-scale conditions in order to determine their efficiency in cadmium removal via ureolysis-driven MICP. They obtained removal rates of 99.95% after incubations of 48 h with spherical $CdCO_3$ crystals. A mine isolate of *Lysinibacillus sphaericus* [269] removed 99.95% of the Cd after 48 h of incubation. Sepúlveda et al. (2021) [270] assessed the potential of the soil-isolated bacteria *Staphylococcus equorum* for MICP of divalent copper removal in comparison with the model microorganism, *S. pasteurii*. Although *S. equorum* does not have as high urease activity as *S. pasteurii*, it has the potential to be used in mining operations when seawater is used because of its halophilic nature. The copper removal was inefficient in both cases because of the formation of copper–ammonia complexes, so optimization of the process is still needed and, therefore, under study. Achal et al. (2011) [271] studied copper removal by *Kocuria flava* CR1, which has high urease activity rates. They achieved a Cu removal of 95%–97% in contaminated soils by the induction of calcite and aragonite precipitation, demonstrating the power of this process. Recently, studies with *Sporosarcina pasteurii* (ATCC11859) and *Terrabacter tumescens* AS.1.2690 achieved removals of up to 99% of Ni, Cu, Pb, Co, Zn and Cd after 48 h incubation by their ureolytic activity [137]. Achal et al. (2012) [272] studied MICP with *Sporosarcina ginsengisoli* CR5 to remove arsenic (III), a highly toxic metalloid, at concentrations lower than 50 mM, proving that the arsenic concentration was higher in the precipitated carbonates after the extraction in the main isomorphs of aragonite, calcite, vaterite and gwihabaite. Puyen et al. (2012) [273] studied Pb and Cu absorption by *Micrococcus luteus* DE2008, which showed its high and low tolerance for Pb and Cu, respectively, and achieved Pb and Cu recoveries of 1,965,000 mg/kg for Pb and 408,000 mg/kg, respectively. In another work [141], batch studies were carried out in order to evaluate the Cd removal by MICP compared to biosorption by *Bacillus* sp. GZ-22 isolated from a mine soil. The Cd removal efficiency was 60.72% at a concentration of 10 mg/kg in contrast to the 56.27% obtained from the biosorption approach, demonstrating that MICP can be a favorable process. Mitchel and Ferris [119,143] investigated MICP in

groundwaters polluted with $Sr^{90}$ and $Sr^{2+}$, demonstrating the capture of $Sr^{2+}$ ions. The removal of chromium by MICP was also demonstrated [274].

The ability of microbial biofilms to remove manganese from mine waters has also been demonstrated [275]. In the case of manganese, processes that effectively remove Mn from mine water by oxidation of soluble $Mn^{2+}$ species at high pH with substrates such as dolomite or limestone have been previously proposed. Nevertheless, in this study, the role of biofilms in Mn removal in reactor-scale experiments with limestone, dolomite, quartzite and magnesite as carbonate sources was investigated. Mn removal was achieved with all of the media (except in quartz). In a study carried out by Marsili et al. (2005) [276], hexavalent uranium was immobilized in reactor-scale experiments by biofilms of the sulfate reducer *Desulfovibrio desulfuricans* G20. The uranium carbonates were observed as mainly attached to the cell membranes and the periplasmic space, with amorphous or poor nanocrystalline structures. The removal efficiency was up to 73% when the reactor was supplied with uranium concentrations of 30 mg/kg. The toxic effects of Pb on the urease activity and the microbial growth were evaluated in the MICP of Pb by *Sporosarcina pasteurii* [277]. The study concluded that the Pb concentration had a marginal effect on microbial growth when this value was above 30 mM.

For bacterial supply in soil, the injection strategy is the most used, promoting $CaCO_3$ precipitation while maintaining ureolytic bacteria in the contaminated soils. Nevertheless, the injection method is difficult to implement in silt, clay and, in general, soils with low hydraulic conductivity. In a recent study [39], an alternative method was considered to mix the calcium salt, nutrients (peptone), *S. pasteurii* cells and the contaminated soil (with heavy metals as $Zn^{2+}$, $Pb^{2+}$ and $Cd^{2+}$). An increase in the $HCO_3^-$ ions and the pH inhibited the competitive effect of $H^+$, which makes the organic/inorganic matter, the Fe-Mn oxide and the contaminating (heavy) metals combine with greater strength, effectively reducing the exchangeable fractions and leading to the precipitation of metal carbonates. Therefore, heavy-metal remediation (bioremediation) by *S. pasteurii* reduced the leaching concentrations of $Pb^{2+}$, $Zn^{2+}$ and $Cd^{2+}$ in the soil through the precipitation of metal carbonates.

### 5.2.3. Plastic Industry

Carbonates can also be applied in the plastic industry, as the new concept of "mineral plastics". This concept emerged from the need to explore new materials independent from nonrenewable resources. For instance, the fabrication of carbonate plastic minerals based on ACC-based hybrid supramolecular hydrogel by mixing $CaCl_2$, $Na_2CO_3$ and polyacrylic acid (PAA) in water has been proposed [278]. The synthesis was based on the idea that acids can help the nucleation of carbonate minerals. A shapeable plastic with a soft but tough structure was obtained, stretchable without shape recovery and able to self-heal rapidly. Such material could easily be colored by thermochromism and could recover its initial properties when made wet. Material analysis showed that the dry gel is composed of ACC nanoparticles physically crosslinked via PAA by Coulomb interactions between the charged groups $COO^-$ and $Ca^{2+}$.

Highly extensible nanocomposite hydrogels have been reported by (i) mixing biopolymers like polyethylene glycol with hydroxyapatite nanoparticles; (ii) ACC-based hybrid gel-like disks, created by mixing $CaCO_3$ within poly(N-isopropylacrylamide) (PNIPAM) microgel dispersant; and (iii) transparent cellulose-ACC films/cellulose-amorphous $CaSO_4$ films [279–282]. Incorporating EICP and MICP as the source of minerals used to create ecoplastics would improve the profit and industrial scale of this process, with very low energy consumption. Some bacterial species are capable of creating their own hybrid biominerals combining organic matrices and amorphous minerals [283,284]. For example, the bacterium *Geobacillus thermoglucosidasius* has been studied to catalyze the formation of fluorescent calcium carbonate from thermophilically composted organic wastes [285]. The resultant calcite presented fluorescence excitation ranges from 260 to 400 nm and emission ranges from 350 to 600 nm. Such a large fluorescence range is in contrast with

those fluorescent calcium carbonates obtained by doping the mineral with fluorescent molecules, making this material interesting as a fluorescent filler without losing fluorescent intensity and stability.

### 5.2.4. Bioconstruction and Cementitious Materials

The construction industry is one of the major economic contributors to society, but it is responsible for very high energy consumption rates as well as huge $CO_2$ emissions. Thus, it is massively practiced, but its contribution to global warming is huge as well. The need to find economical green alternatives becomes more a reality day by day. Cements are usually used as typical binding powder that makes possible the creation of concrete and mortar, with concrete being one of the most used materials on Earth [286]. Biocementation emerges as an ecofriendly alternative in construction materials, in which microorganisms precipitate calcium carbonate with construction purposes. This technique is based on the idea that the binding strength of the precipitated crystals is highly dependent on the rate of carbonate formation, and, under suitable conditions, it is possible to control the reaction to generate hard-binding calcite biocement.

Several approaches for MICP have been proposed, which mainly consist of the injection of the microorganism, an adequate media with nutrients and the calcium salt inside the porosities of the limestone surfaces of the buildings, concrete and mortar structures, as exemplified in Figure 6 [287]. The final features of the biocement depend on the microorganism (*Sporosarcina pasteurii* or *Bacillus mucilaginous*, among other ureolytic or nonureolytic microorganisms), and the type of calcium source (calcium acetate, calcium chloride, calcium nitrate, etc.) in order to obtain a relatively stable polymorph [288]. The most studied microorganism for biocementation is *S. pasteurii* because of the high urease activity rates and its ability to form vaterite and calcite isomorphs. *S. pasteurii* has been shown to produce cement with a reduced porosity that presents increased compressive strength and reduced water adsorption, features that enhance the durability of the material [289]. The enhancement of the MICP properties driven by *S. pasteurii* has been used in sand and mortar cementation [290–293]. It has been shown that *S. pasteurii* improved sand consolidation by decreasing by 50% the porosity and by 90% the permeability of the sands tested [294]. To remediate cracks in the concrete, MICP was tested with other microorganisms, such as *B. pasteurii*, *B sphaericus* or *Bacillus* sp. CT5 [295–297]. MICP has also been used to reduce the permeability of stone surfaces to increase the durability of the stones. This is an application to enhance the protection and robustness of limestone buildings and decrease the stone erodibility by environmental factors. A study to evaluate the reduction of the porosity of limestone buildings using *Micrococcus* sp. and *B. subtilis* observed a reduction of approximately 60% of the porosity but evidenced some negative side effects, such as the formation of stained patches due to the growth of microfungi due to the rich medium needed for the growth of bacteria [298]. Studies with *B. sphaericus* and *Bacillus lentus* on limestone cubes showed increases in the pH, calcite precipitation over the cubes with the production of EPS and biofilm, and urease activity [299].

MICP has been proposed for the restoration of historical buildings deteriorated by atmospheric and biological causes, among which fungal patinas are very frequent [300,301]. Bacteria, such as *Desulfovibrio desulfuricans* and *D. vulgaris*, have been used to dissolve sulfate black crusts to restore polluted buildings or stone artworks [302].

Parallel to MICP for biocementation purposes or restoration and remediation of damaged stone materials, EICP is under development. EICP has a promising potential for dust control in cities [303] as well as for improving the consolidation of sands and soils [304]. In a study performed at the laboratory and field scales on EICP by ureases, in order to enhance the efficiency for dust control and increase soil resistance against rainfall-induced erosion, the strongest erosion resistances were achieved when the soil was treated with the most stable $CaCO_3$ precipitate in combination with polyvinyl acetate (PVAc) [305]. Treatments with higher concentrations of urease enhanced the formation of ACC and aragonite in higher proportions, while the use of low urease concentrations led to

the formation of calcite. Such promising results suggest that research on EICP should be increased for applications in the field of biocementation.

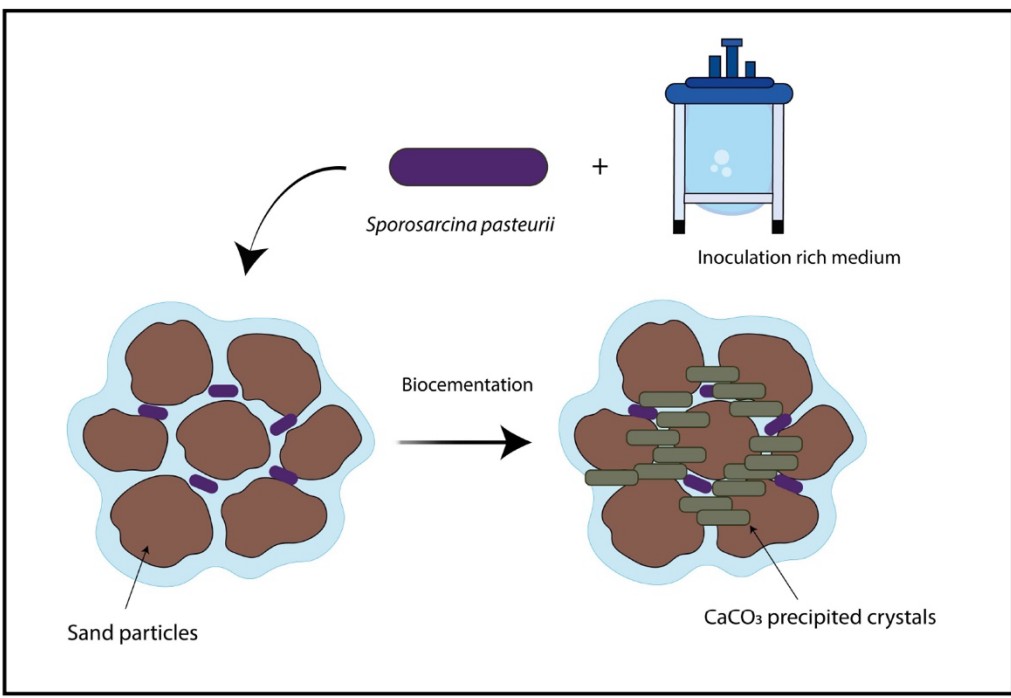

**Figure 6.** Simplification of the biocementation via the MICP process. The figure shows the main scheme of the process of injection of the microorganisms within an inoculation-rich medium inside the pores of the cementitious material (i.e., sand grains). In one or two days, depending on the microorganism inoculated, the precipitation of $CaCO_3$ crystals would enhance the adhesion and compaction of the material by occupying the pore's space, enhancing the resistance properties of the cement.

## 6. Conclusions

Many different microorganisms and higher organisms can induce and determine MICP through the production of EPS, enzymes, amino acids and other organic metabolites that by their interactions determine the type and rate of the process. Many studies have focused on the molecular routes and metabolic pathways activated by these organisms that determine and regulate the precipitation process, suggesting that the final properties of the minerals obtained are genetically regulated and control of the process is achievable. Understanding the parameters for the synthesis of the organic molecules involved in MICP and EICP might shed light on developing fine-tune control of these processes for biotechnological applications. Carbonates are important molecules for carbon stock and for potentially regulating the current trends in $CO_2$ accumulation in the atmosphere. Moreover, carbonates are very useful minerals in several engineering, medical engineering and geological applications. Although many questions remain to be answered regarding the role of microorganisms in the production and turnover of carbonates, such as the kind of microorganisms able to induce the process and their ecology and interactions; the role of the environmental factors involved; or the role and mechanisms of the biological molecules (EPS, proteins, enzymes, organic cofactors, organic acids, etc.) involved, current research has shaped the biologically mediated precipitation and rearrangements of carbonates as a world-scale process that has very great interest for the protection of the environment, regulation of the climate and application in many (geo)biotechnological problems that may contribute to the understanding of the past of our planet and the search for life on other planets.

**Author Contributions:** M.S.-R. and D.D. performed the conception of the study. M.S.-R. designed the study, assessed the results, advised in the practical execution and manuscript writing. D.D. and V.C.V. revised and edited the manuscript. C.A. advised in the practical execution and manuscript writing. A.R.-F. prepared the original draft, conducted the practical work and assessed the results. All authors have read and agreed to the published version of the manuscript.

**Funding:** This research was funded by the Competitive Research Grant (CRG) of the King Abdullah University of Science and Technology (KAUST)—grant OSR# 4097—MiMiC project.

**Acknowledgments:** A.R.-F. thanks M.S.-R. and C.A. for their constant support, patience, availability, and kindness in relation to all her doubts and concerns during the elaboration of this review. A.R.-F. also wants to thank D.D. and V.C.V. for their suggestions and corrections, which helped her to improve the review. Finally, A.R.-F. thanks V.V., the King Abdullah University of Science and Technology (KAUST), and its Visiting Student Research Program (VRSP) for giving her the opportunity to work within the frame of the MIMIC project and for the trust deposited in her.

**Conflicts of Interest:** The authors declare no conflict of interest.

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
