# Peer review of "The Role of Microorganisms in the Nucleation of Carbonates, Environmental Implications and Applications"

_minerals, doi:10.3390/min12121562_

Round 1
Reviewer 1 Report
The manuscript is well prepared. They main idea of this paper is enlightened with plenty of previous research.
Please revise the manuscript according to the comments below-
1. Use SI units. Convert all mg/L, mg/g to mg/kg.
2. References must be numbered in order of appearance in the text. You used [54] before [52], [53].
Author Response
Dear Ms. Astrid Yang,
We thank you for considering our manuscript for publication in Minerals. We thank you for your advice on our manuscript, and we hereby to re-submit the revised version of the manuscript entitled “The role of microorganisms in the nucleation of carbonates, environmental implications, and applications” as requested for further consideration.
We have carefully addressed all the comments, provided by the three reviewers, within the main text of the manuscript. We would also like to thank the positive comments, feedback, and constructive comments provided by reviewers #1, #2 and #3 that helped to improve this version of the manuscript. Below you can find the point-by-point response.
We hope that this new revised version of the manuscript, based on the suggestions/edits/changes made by the three reviewers, meets your expectations to be published in Minerals.
I thank you in advance and, please, don’t hesitate to contact me whether you need further information or have any questions about the present reviewed manuscript.
Sincerely yours,
Mónica Sánchez-Román
On behalf of the authors

Reviewer 2 Report
Dear authors,
I find really interesting the manuscript and about a topic that needed to be reviewed, highlighting the importance of microorganisms in the formation and precipitation of carbonates. I suggest however some changes in the structure, since in my view there is too much information about the different biological polymers that exist, something already known and too detailed for this type of MDPI journal that is Minerals. Thus, I would make sure to focus on the title of the manuscript throughout the text. I suggest to change the title to "The role of microorganisms in the nucleation of carbonates", shorter, catchy and self-descriptive.
Some specific questions:
1. The definitions of biological influenced mineralization, biological induced mineralization and biological controlled mineralization, are you introducing them for the first time or have you read it before? I have always seen other definitions like:
a. Active/direct microbial biomineralization or biological mineralization: Minerals precipitate by microbial metabolisms (i.e. sulfate reduction).
b. Passive/indirect microbial biomineralization or biological mineralization: minerals precipitate by the presence of microorganisms in the environment and the molecules and compounds they may have in their cell-wall or EPS, acting as passive nucleation sites.
Some suggestions:
· Remove section 2 (Biological polymers) and incorporate relevant information form this section to section 3 (Microorganisms as producers of organic compounds).
· Reduce drastically the size of the manuscript. I suggest something like:
o 1. Introduction
o 2. (Micro)biological compounds relevant for carbonate precipitation
o 3. Types of microbial biomineralizations.
o 4. Polymeric biological substances associated with carbonates
o 5. Implications and applications

Author Response

(The authors gave the same response as above.)

Reviewer 3 Report
This review by Robles-Fernández et al. attempt to summarize the effect of organic compound from microorganisms on the formation of carbonate. The basic misunderstanding on the concepts of biomineralization and aqueous chemistry as well as ignoring significant works in the field of biomineralization and carbonate especially from Dove, De Yereo, and Gilbert’s groups make this review not incomplete. Throughout the article, the authors are citing inappropriate references that do not supporting their claims. It appears that they do not have an explicit understanding of the references. Additionally, the title imply this review will be focus on role of microorganism on carbonate formation, significant length was focus on technology implications. The authors also failed to explain how these organic compounds actually promote carbonate nucleation from a detail molecular/atomic/thermodynamic manner. Therefore, I do not think this article is suitable for publication.
Detail comments:
Line 23: Please rephrase “power controlling” this is too colloquial.
Line 24: I do not understand what the authors mean by “highlighted in the spotlight”. Please consider rewriting this sentence.
Line 39: I think the authors need to cite MTB reference here for oxides.
Line 43: This is an inappropriate citation as Dupraz et al. (2009) is talking only about carbonate in microbial mats but the concept of biomineralization is significantly broader. The idea of biomineralization was proposed much earlier.
Line 49-51: These two reference are inappropriately cited as there are talking about very specific aspects of biomineralization process. Weiner and Dove (2003) provides a much more appropriate reference for BIM and BCM.
Line 53: Please change “of” to “on”.
Line 54-56: This is not the definition of biomineralization and the references are also not appropriate!!! There are much more to biomineralization than just microbial mat such as corals, bivalves, crustaceans and more!
Line 59: References 7, 22, 23, and 24 are talking about very specific aspects of microbial influence on dolomite formation and therefore are not correct to cite them here. Gilbert et al. (2022), Gilbert et al. (2019), and De Yoreo et al. (2015) are much better suited references.
Line 73: “With respect to”
Line 97-99: Reference for this sentence.
Line 120: The authors changes the definition of EPS here from their previous text. Although the two terms are interchangeable, please be consistent in your work on terminology.
Line 137-139: This sentence is fragmented. Please rewrite.
Line 148: Please separate the paragraph from the sentence starting with “Apart” since the topic has changed.
Line 157-159 Again, this reference is talking about polysaccharides on carbonate formation. If the authors want to make a broader claim, please use a broader topic reference.
Line 167-171: The sentence is awkward. Please rewrite it.
Figure 2: This is not the first paper that mention this concept. Therefore it is necessary to cite appropriate reference.
Line 209: Again, these are not appropriate references!!
Line 228-229: This is a false statement. There are many examples of biological induced and biological controlled calcium carbonate!!! For example, mollusks will CONTROL their calcium carbonate growth, morphology, and composition and nacre.
Line 229-231: This is a claim without appropriate support. Also, the reference is inappropriately cited and the authors (Gallagher) did not make the claim that MICP is the most suitable mechanism for carbonate precipitation.
Line 235-236: Incomplete sentence.
Line 242-243: This sentence is incorrect. First, the author need to spell out abbreviation for IAP , which is ion activity product. Second, IAP is not the product of concentration of calcium and carbonate. IAP is the ACTIVITY (which is the effective concentration that are different from the actual concentration) product of calcium and carbonate.
Line 243-245: This is also an incorrect sentence. When IAP = Ksp, SI = 0 which is how log works. Therefore, when IAP > Ksp, SI > 0 but not necessarily closer to 1. It could be significantly larger than 1 or close to 0.
Equation 2: This is not balanced. Please include hydrogen in the equation. Also, if the equation is reversible, please use a double arrow instead of quoting.
Line 251-253: I do not understand what the authors meant by “alkalinity condition”. Do you mean alkaline?
Line 256: “alkalinity” not “Alkalinity”.
Line 267-269: Fragmented sentence. Missing reference.
Line 269: “their calcium concentration”.
Line 271-275: This sentence is fragmented and awkward. Please rewrite.
Figure 3: This is not the first paper that mention this concept. Therefore, it is necessary to cite appropriate reference.
Line 301: “cyanobacteria” not “Cyanobacteria”.
Line 308: “thanks to CA” is too colloquial. Please be formal in scientific writing.
Line313: This is inappropriately cited. The reference is not focus on SRB but briefly mention it. Please look into the reference and cite original work that describe dissimilatory sulfate reduction.
Line 314: This is also not an appropriate reference. The reference focus on the role of SRB on dolomite formation not the function role of SRB to provide bicarbonate. Jorgensen (1977) is a more suitable reference.
Line 329-334: This is a wordy and incomplete sentence. Please rewrite.
Line 334-336: This sentence is really confusion especially what do the authors meant by “Ca2+ added to the higher rate of urease”. Please rewrite.
Line 342: I do not follow the authors logic here. Since both side suggest higher than 8 pH, why use “but” to connect.
Line 343-347: Eq. 3-6 produce 2 H+ and 1 OH- which should decrease the pH. Also, please reword “releasing”.
Line 365: Please spell out EICP.
Line 430: Arsenic in nature are usually +3 or +5 charged not +2.
Line 1167: Conclusion?
Author Response

(The authors gave the same response as above.)

Round 2
Reviewer 3 Report
I think the authors have significantly improved the manuscript from last time. There are still some issues that I have described in detailed comments that need to be addressed. One thing I did not point out in detailed comments is the mixed usage of the terms BIC and MICP, especially when they appear in two consecutive sentences, which might confuse readers. It would be better to use the terms more consistently and carefully. I would suggest acceptance after some revision.
Detail comments:
Line 32: The increased nucleation site is independent from oversaturation. Please rephrase.
Line 33: “to an excess of the solubility product” is redundant as it is already implied in the oversaturation.
Line 41-43: References for shells for crustaceans' exoskeleton (Hild et al., 2008, J. Stur. Biol.; Li et al., 2020, Nature Comm.) and bones for mammals and birds are missing here. Bivalve is one kind of mollusk. Ref 25 is for eggs not bones.
Line 47-49: Full names for BIM and BCM are incorrectly presented. Weiner used “biological induced/controlled biomineralization”.
Line 73-74: This statement is false. Chemical precipitation of minerals are not necessarily a slow process. For example, mineral precipitating in hypersaline lakes have a extremely fast rate.
Line 84: Spell out MICP. This is the first time the term appears.
Line 87-91: This is a very confusing sentence. Readers shouldn’t need to guess the relationship between MICP and EICP.
Line 132-133: Reference is needed here.
Line 141-143: This is an incomplete sentence.
Line 156-157: Reference is needed here.
Line 158-159: Reference is needed here. Based on previous work, EPS does not selectively concentrate Mg but allowing Mg to dehydrate so they can be incorporated into carbonate structure. Please correct this.
Line 160-161: Reference is needed here.
Line 193-196: This sentence is a bit repetitive and should be combined with the previous paragraph.
Line 242: If IAP is higher than Ksp, the saturation index is positive not increased.
Line 500-503: I should point out that alcohols could also affect crystal morphologies, polymorphs (Seo et al., 2005, Journal of Crystal Growth; Sand et al. 2012, Crystal Growth and Design), and Mg composition (Fang et al., 2021, ACS Omega).
Author Response
Dear Ms. Astrid Yang,
We thank you for considering our manuscript for publication in Minerals. We thank you for your advice on our manuscript, and we hereby resubmit the revised version of the manuscript entitled “The role of microorganisms in the nucleation of carbonates, environmental implications and applications” as requested for further consideration to be published in Minerals.
We have carefully addressed all the new comments, provided by reviewer #3, within the main text of the manuscript. We would also like to thank the positive comments, feedback, and constructive comments provided by reviewer #3 that helped to improve this new revised version of the manuscript. Below you can find the point-by-point response to reviewer #3. Please find it attached.
We hope that this new revised version of the manuscript, based on the suggestions/edits/changes and feedback from the reviewer #3, meets your expectations to be published in Minerals.
I thank you in advance and, please, don’t hesitate to contact me whether you need further information or have any questions about the present new version of the reviewed manuscript.
Sincerely yours,
Mónica Sánchez-Román
On behalf of the authors
